# Beyond Confidence: Adaptive and Coherent Decoding for Diffusion Language Models

**Kecheng Chen** [* 1]   **Ziru Liu** [* 2]   **Xijia Tao** [3]   **Hui Liu** [1]   **Xinyu Fu** [2]   **Suiyun Zhang** [2]   **Dandan Tu** [2]   **Lingpeng Kong** [3]   **Rui Liu** [2]   **Haoliang Li** [1]

## Abstract

Diffusion Language Models (DLMs) have recently achieved significant success due to their any-order generation capabilities. However, existing inference methods typically rely on local, immediate-step metrics, such as confidence or entropy, which inherently lack a more reliable perspective, leading to sub-optimal generation quality. To address this, we propose **C**oherent **C**ontextual **D**ecoding (**CCD**), a novel inference framework built upon two core innovations. First, CCD bypasses the potential bias of the single context to leverage historical contexts for approximating the marginal distribution of token prediction, leading to better sequence coherence and the early rejection of sub-optimal paths. More importantly, we demonstrate that this mechanism is theoretically equivalent to modeling the consistency of historical steps via the conditional mutual information between contexts and token predictions. Finally, CCD achieves significantly milder performance degradation under highly parallel decoding scenarios compared to baselines. Empirically, our method achieves a simultaneous enhancement in both inference speed and performance across diverse benchmarks on Dream and LLaDA. The code and project page are publicly available at `https://tonyckc.github.io/CCD-DLM-Project/`.

## 1. Introduction

Diffusion Language Models (DLMs) have emerged as a compelling alternative (DeepMind, 2025; Inception Labs, 2025) of current autoregressive language models (OpenAI, 2025; DeepSeek-AI et al., 2025). Unlike autoregressive models restricted to unidirectional contexts, DLMs leverage bidirectional contexts, incorporating information from all positions simultaneously. This global perspective allows DLMs to potentially achieve superior global awareness and long-term planning capabilities. With recent scaling to the 7–8 billion parameter regime (e.g., the Dream (Ye et al., 2025) and LLaDA (Nie et al., 2025)), masked diffusion mechanisms have proven capable of matching the performance of their autoregressive counterparts while exhibiting significantly better long-term planning. However, in contrast to the rapid advancements in training strategies, the inference (sampling or decoding) procedure of DLMs remains relatively underexplored. Specifically, DLMs iteratively decode semantically meaningful tokens from input mask tokens based on the previously decoded context. Existing methods typically leverage the predictive distribution at the current step to select the top-$k$ or most certain tokens for unmasking (Ye et al., 2025). To maintain generation performance, the decoding steps are usually proportional to the sequence length (Nie et al., 2025), *i.e.,* a uniform sampling scheme with a decoding budget of 1 in each step.

Despite their empirical success, we argue that current single-step sampling procedures suffer from three critical limitations: (i) **Susceptibility to Local Optima:** Whether relying on entropy or maximum probability, single-step certainty-based schemes are prone to settling for local optima. This issue is exacerbated by the tendency of modern LLMs to exhibit over-confident even in error predictions (Sun et al., 2025). (ii) **Lack of Theoretical Grounding:** Current sampling procedures lack a direct theoretical bridge to the sampling error rate, making it difficult to govern the decoding process for controllable performance guarantees. (iii) **Catastrophic Degradation on Highly Parallel Decoding Scenarios:** Existing DLMs with single-step sampling procedures usually suffer from nonnegligible performance degradation on the increased decoding budget for each step (Wu et al.,

---
*Equal contribution  [1]City University of Hong Kong, Hong Kong SAR, China [2]Huawei Research [3]The University of Hong Kong, Hong Kong SAR, China. Correspondence to: Dandan Tu <tudandan@huawei.com>, Rui Liu <liu.rui2@huawei.com>, Haoliang Li <haoliang.li@cityu.edu.hk>.

*Proceedings of the $43^{rd}$ International Conference on Machine Learning*, Seoul, South Korea. PMLR 306, 2026. Copyright 2026 by the author(s).

2025), which impedes the pursuit of inference efficiency.

To address these challenges collectively, we introduce **C**oherent **C**ontextual **D**ecoding (**CCD**), a novel inference framework that redefines DLM inference as a contextual consistency-aware, multi-step decoding process. CCD adopts a new perspective by modeling the consistency of historical steps using the conditional mutual information between the context and token predictions. This provides an effective connection with the theoretical upper bound of sampling error rate in DLMs. Furthermore, we extend CCD into an adaptive sampling version (CCD-DS) that dynamically increases the decoding budget for each step. As the mechanism of CCD naturally prioritizes those tokens that exhibit high *confidence* and *predictive consistency* across steps, CCD-DS enjoys efficiency gains without obvious performance degradation. Our contributions are as follows:

- (Section 2.1) We provide a thorough analysis demonstrating that existing sampling procedures utilized the decoding-time single-step predictive distribution as a surrogate of the target distribution is suboptimal.

- (Section 3.1 & 3.2) Theoretically, we introduce the evolving context of DLMs as an auxiliary variable to approximate the target distribution, which connects to the single-step distribution and governs the sampling error bound.

- (Section 3.3) Empirically, we propose CCD (as illustrated in Fig. 2) and its adaptive sampling budget extension as a training-free realization of context-driven target distribution approximation.

- (Section 4.2) Experimentally, CCD achieves performance gains under a uniform sampling scheme (Table 1) and milder performance degradation under highly parallel decoding scenarios compared with existing counterparts (Tables 2 and 3). CCD-DS improves the quality of generation trajectories while accelerating the sampling process.

## 2. Preliminary and Motivation

**Notation.** Let $\overline{\mathbb{X}} = \mathbb{X} \cup \{\mathrm{M}\}$ be the extended vocabulary of a finite-size vocabulary $\mathbb{X}$, where $\mathrm{M}$ denotes an absorbing mask token. Let $\mathbf{s} \in \mathbb{X}^L$ be a prompt sequence of length $L$. Let $\mathbf{x}_t = \{x_{t,1}, \ldots, x_{t,N}\} \in \overline{\mathbb{X}}^N$ denote the response sequence at time step $t$, where $t \in \{0, \ldots, T\}$ indicates the diffusion step. We adhere to the standard diffusion formulation where the process evolves from a fully masked state $\mathbf{x}_T = \{\mathrm{M}\}^N$ to a fully unmasked data state $\mathbf{x}_0 \in \mathbb{X}^N$. We denote the concatenation of the prompt and response at step $t$ as $[\mathbf{s}; \mathbf{x}_t]$. Hereafter, $\hat{\mathbf{x}}_t$ and $\overline{\mathbf{x}}_t$ denote the predicted and true response sequences at iteration $t$, respectively.

**Preliminary for Sampling Procedure of DLMs.** A DLM approximates the reverse process as a sequence-to-sequence

mapping parameterized by $\theta$, which outputs a predictive distribution $\hat{p}_\theta$. The model progressively decodes mask tokens starting from $\mathbf{x}_T$. At each iteration, the model takes the fixed prompt $\mathbf{s}$ and the state from the previous iteration $\mathbf{x}_{t+1}$ to estimate the clean data $\mathbf{x}_0$.

Typically, the generation budget $b_t$ (the number of tokens to unmask at step $t$) is uniform across iterations (see the dashed curve of Fig. 3(b) in Appendix), such that $b_t \approx N/T$ for all $t \in [T]$. Under this setup, the most widely adopted sampling schedule[1] utilizes the predictive distribution to select the top-$b_t$ most certain tokens for decoding. We employ the negative entropy of the predictive distribution as a generic *surrogate* for confidence[2]. The sampling procedure[3] from $\mathbf{x}_{t+1}$ to $\mathbf{x}_t$ is formulated as:

$$
x_{t,i} = \begin{cases} \arg\max\limits_{k \in \mathbb{X}} p_{t,i}^k & \text{if } i \in \mathcal{J}_t \\ x_{t+1,i} & \text{otherwise} \end{cases}, \tag{1}
$$

where the components are defined as follows: $p_{t,i} = \hat{p}_\theta(\cdot \mid x_{t+1,i} = \mathrm{M}, \mathbf{c}_{t,i}, \mathbf{s})$ is the predictive distribution at position $i$, given the prompt $\mathbf{s}$ and context $\mathbf{c}_{t,i}$. Here, $\mathbf{c}_{t,i}$ denotes the set of unmasked tokens that serve as the context for step $t$ for position $i$. $\mathcal{J}_t \subset \mathcal{K}_t$ is the set of indices selected for decoding, determined by maximizing certainty:

$$
\mathcal{J}_t = \arg\max\limits_{\mathcal{S} \subset \mathcal{K}_t, |\mathcal{S}| = b_t} \sum_{i \in \mathcal{S}} -H(p_{t,i}), \tag{2}
$$

where $\mathcal{K}_t = \{i \mid x_{t+1,i} = \mathrm{M}\}$ is the set of indices currently masked. Here, $H(p_{t,i}) = -\sum_{k \in \mathbb{X}}(p_{t,i}^k \log p_{t,i}^k)$ denotes the Shannon entropy.

As illustrated in Eq. (1), existing sampling strategies predominantly rely on the predictive distribution of the DLM at a specific position $i$ given the current state, denoted as $\hat{p}_\theta(\cdot \mid x_{t+1,i}, \mathbf{c}_{t,i}, \mathbf{s})$. This distribution not only determines the entropy statistics for the sampling index set $\mathcal{J}_t$, but also dictates the specific token values via probability mass. Since the input token $x_{t+1,i}$ is a uniform mask token for all indices in $\mathcal{K}_t$ and the prompt $\mathbf{s}$ is fixed, the predictive distribution is primarily conditioned on the variable context $\mathbf{c}_{t,i}$. Consequently, we refer to $\hat{p}_\theta(\cdot \mid x_{t+1,i}, \mathbf{c}_{t,i}, \mathbf{s})$ as the *single-step predictive distribution* hereafter.

### 2.1. Motivation

To quantify the quality of the sampling procedure governed by this single-step predictive distribution, we introduce the Token Error Rate (TER) (Feng et al., 2025) as follows.

---

[1]Random selection is feasible but yields degraded performance.

[2]Negative entropy serves as a smooth approximation to confidence-based metrics (e.g., margin confidence), particularly given that LLM distributions tend to be highly overconfident

[3]We formulate this using deterministically greedy sampling ($\arg\max$), but it can be extended to categorical sampling.

**Definition 2.1** (**Token Error Rate for Conditional Generation**) *Given a target data distribution $p(\cdot \mid \mathbf{s})$ and the model's predictive distribution $\hat{p}(\cdot \mid \mathbf{s})$, the TER is defined as the expected negative log-likelihood per token:*

$$\log_2 \text{TER}(\hat{p}(\cdot \mid \mathbf{s}); p(\cdot \mid \mathbf{s})) \triangleq -\frac{1}{N}\mathbb{E}_{\mathbf{x}\sim p(\cdot\mid\mathbf{s})}\big[\log \hat{p}(\mathbf{x} \mid \mathbf{s})\big],$$

where $N$ is the sequence length. This metric captures the alignment between the generated and target distributions. To analyze this at the token level, we decompose the TER using the decoding-time single-step predictive distribution $\hat{p}_\theta(x_i \mid \mathbf{c}_{.,i}, \mathbf{s})$, where position $i$ is decoded into $x_i$ using a specific decoding-time context $\mathbf{c}_{.,i}$. Assuming the target distribution factorizes independently across positions given the prompt for analytical tractability, we derive $\log_2 \text{TER}(\hat{p}(\cdot\mid\mathbf{s}); p(\cdot\mid\mathbf{s})) = -\frac{1}{N}\sum_{i=1}^N \mathbb{E}_{x_i\sim p(x_i\mid\mathbf{s})}\left[\log \hat{p}_\theta(x_i\mid\mathbf{c}_{.,i},\mathbf{s})\right]$. Minimizing this quantity is equivalent to minimizing the KL divergence between the target and predictive distributions, as the entropy of the true data distribution is constant. Thus, the optimization objective can be formulated as:

$$\log_2 \text{TER}(\hat{p}(\cdot\mid\mathbf{s}); p(\cdot\mid\mathbf{s})) \propto \frac{1}{N}\sum_{i=1}^N D_{\text{KL}}(p(x_i\mid\mathbf{s})\|\hat{p}_\theta), \quad (3)$$

where $\hat{p}_\theta = \hat{p}_\theta(x_i\mid\mathbf{c}_{.,i},\mathbf{s})$. Based on this analysis, we identify two critical limitations in existing sampling procedures.

- **Implicit approximation of target distribution using decoding-time single-step predictive distribution:** Existing sampling procedures implicitly assume that decoding-time single-step predictive distribution is a reliable surrogate for the target distribution, *i.e.,* $p(x_i\mid\mathbf{s}) \approx \hat{p}_\theta(x_i\mid\mathbf{c}_{.,i},\mathbf{s})$. However, such an approximation may be vulnerable when the decoding-time context $\mathbf{c}_{.,i}$ in the current step is itself inaccurate, resulting in an unreliable and degraded predictive distribution. Correct sampling results cannot be guaranteed.

- **The TER for conditional generation cannot be governed appropriately.** Once an incorrect predictive distribution arising from an ineffective decoding-time context is considered, there is a large distribution discrepancy (*i.e.,* $D_{\text{KL}}$ is large) between the single-step predictive distribution and the target distribution in Eq. (3). An appropriate level of TER thus cannot be guaranteed.

## 3. Methodology

### 3.1. Approximation of Target Distribution

Here, a straightforward approach would be to compute the target marginal distribution $p(x_i \mid \mathbf{s})$ directly, thereby bypassing the bias of any single context. While computing the true marginal is intractable, we propose approximating it by exploiting the intrinsic iterative nature of DLMs.

> **Core insight.** Unlike autoregressive models, a DLM generates multiple predictive distributions for a masked position $i$ across the decoding trajectory $(T \to t)$ before that position is unmasked. These intermediate predictions are conditioned on evolving contexts. We can approximate the true marginal distribution by integrating over these contexts.

Mathematically, for any joint distribution, we can express the marginal distribution by introducing an auxiliary variable and integrating it out, *i.e.,* $p(x_i \mid \mathbf{s}) =$

$$\int p(x_i, \mathbf{c} \mid \mathbf{s})d\mathbf{c} = \int p(x_i \mid \mathbf{c}, \mathbf{s})p(\mathbf{c} \mid \mathbf{s})d\mathbf{c}. \quad (4)$$

In the context of diffusion, the decoding trajectory provides a sequence of context states. We treat the sequence of contexts $\{\mathbf{c}_T, \ldots, \mathbf{c}_t\}$ generated by the model as samples from the context distribution $p(\mathbf{c} \mid \mathbf{s})$. We approximate the integral via an ensemble average over the trajectory, *i.e.,* $p(x_i \mid \mathbf{s}) \approx \overline{p}(x_i \mid \mathbf{s}) \triangleq$

$$\frac{1}{T-t+1}\sum_{k=0}^{T-t} \hat{p}_\theta(x_i \mid x_{T-k,i} = \text{M}, \mathbf{c}_{T-k,i}, \mathbf{s}), \quad (5)$$

where $\overline{p}(x_i \mid \mathbf{s})$ is the approximated target distribution. This effectively performs a Monte Carlo (MC) estimation using $T - t$ correlated samples from the decoding process. Then, one can integrate the approximated target distribution into existing sampling procedures in Eq. (1), where

$$p_{t,i} = \overline{p}(x_i \mid \mathbf{s}), \quad \mathcal{J}_t = \underset{\mathcal{S}\subset\mathcal{K}_t, |\mathcal{S}|=b_t}{\arg\max}\sum_{i\in\mathcal{S}}-H(p_{t,i}). \quad (6)$$

Therefore, from the perspective of the overall decoding trajectory, Eq. (6) actually prioritizes those tokens that exhibit high *confidence* and *predictive consistency* across steps. In other words, if the distribution of a token fluctuates dramatically due to the changing contexts, the entropy of the averaged distribution $\overline{p}$ will be high, deprioritizing its selection. Conversely, tokens that are robust to context variations will be selected earlier. Instead, Eq. (1) only considers tokens with high confidence in the current step.

### 3.2. Theoretical Analysis

We now demonstrate that our approximated target distribution not only connects theoretically to the existing single-step predictive distribution, but also implicitly governs the sampling error bound.

**Proposition 3.1** (**Connection via Mutual Information**). *The approximated target distribution obtained by marginal-*

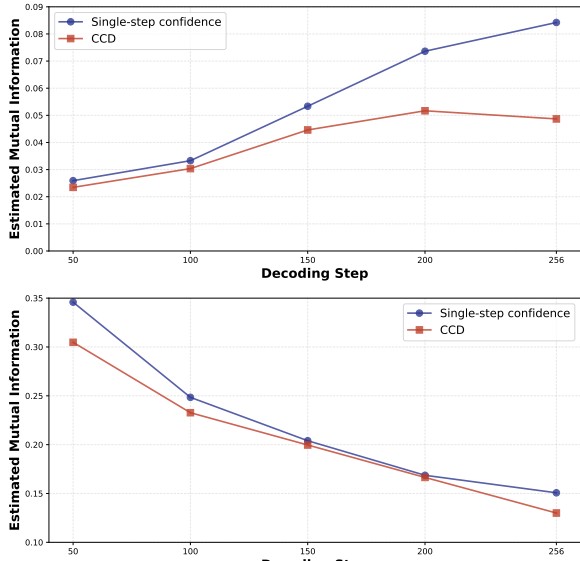

Figure 1. This analysis investigates the estimated average MI ($I(x_i; \mathbf{c}|\mathbf{s}) \approx H(x_i|\mathbf{s}) - H(x_i|\mathbf{c}_{\cdot,i}, \mathbf{s})$ ) on different decoding steps using the Dream model on the Trip (top) and GSM8K (bottom) benchmarks. The sequence length is set to 256. We illustrate the change of the estimated MI throughout the decoding process by randomly sampling 16 generative sequences and calculating the average MI of each step. Note that sampling difficult tokens (high MI) usually exist in the tail of the sequence on the Trip (see the example in Appendix Table 10), while the early stage of the reasoning process on the GSM8K is usually semantically vague (see the example in Appendix Table 8) with potential high MI.

izing over contexts is related to the decoding-time single-step predictive distribution through the conditional mutual information (MI) between contexts and token predictions:

$$H(x_i|\mathbf{s}) = H(x_i|\mathbf{c}, \mathbf{s}) + I(x_i; \mathbf{c}|\mathbf{s})$$
$$\propto H(x_i|\mathbf{c}_{\cdot,i}, \mathbf{s}) + I(x_i; \mathbf{c}|\mathbf{s}) \quad (7)$$

where $I(x_i; \mathbf{c}|\mathbf{s})$ is the conditional mutual information between $x_i$ and $\mathbf{c}$ given $\mathbf{s}$, and $\mathbf{c}_{\cdot,i}$ is the decoding-time context.

Detailed proofs are provided in B.1. Prop. 3.1 uncovers two advantages of the approximated target distribution over contexts for sampling procedures as follows.

(i) **Flexible connection with the existing sampling procedure:** Our approach exhibits flexibility—when $x_i$ and $\mathbf{c}$ are conditionally independent given $\mathbf{s}$, we have $I(x_i; \mathbf{c}|\mathbf{s}) = 0$, and our method naturally degrades to the existing single-context method with $H(x_i|\mathbf{s}) = H(x_i|\mathbf{c}_{\cdot,i}, \mathbf{s})$.

(ii) **Additional consideration of statistical dependencies:** When contexts are informative about $x_i$, our method captures additional statistical dependencies through the $I(x_i; \mathbf{c}|\mathbf{s})$ term, leading to better-calibrated predictions.

Moreover, we uncover that our proposed approximated target distribution over contexts can directly govern sampling

error upper bound based on Proposition 3.1. We first recall the sampling error upper bound of DLM for a general non-conditional sampling (generation) process:

**Lemma 3.2.** *(Li & Cai, 2026) For a well-trained DLM with any uniform mask size schedule, let $\overline{\mathbf{x}} \in \mathbb{X}^N$ be the ground-truth token sequence, and let $x_i$ and $x_{-i}$ be each token at position $i$ and the rest of the sequence, the sampling error of the final output sequence $\hat{\mathbf{x}} \in \mathbb{X}^N$ of the sampling procedure satisfies*

$$\mathbb{E}\big[\mathsf{KL}(p(\overline{\mathbf{x}}) \parallel p(\hat{\mathbf{x}}))\big] \leq \frac{G}{T} \sum_{i=1}^{N} I(\overline{x}_i; \overline{x}_{-i}) + \varepsilon_{train}. \quad (8)$$

*Here, the expectation is taken over all possible mask sets throughout sampling processes. $G$ is a constant depending on the mask schedule, and $T$ is the total number of decoding iterations. The term $\varepsilon_{train}$ represents the inherent training error of the mask predictor. $I(\overline{x}_i; \overline{x}_{-i})$ represents the mutual information between each token and the rest of the sequence.*

Note that the bound reveals that the potential sampling error depends on the intrinsic statistical coupling between tokens in the sequence. Specifically, sequences with higher mutual information between tokens exhibit larger potential sampling errors. Based on this well-constructed sampling error upper bound, we can make the following proposition for conditional generation:

**Proposition 3.3** (**Sampling Error Governance via Marginalized Contexts**). *Under the conditional generation with the $\overline{p}(x_i|\mathbf{s})$ obtained by marginalizing over contexts, as $t \to 0$, the sampling error bound is governed by*

$$\mathbb{E}\big[\mathsf{KL}(p(\overline{\mathbf{x}}|\mathbf{s}) \parallel p(\hat{\mathbf{x}}|\mathbf{s}))\big] \leq \frac{G}{T} \sum_{i=1}^{N} I(\overline{x}_i; \overline{x}_{-i}|\mathbf{s}) + \varepsilon_{train} \approx$$

$$\frac{G}{T} \sum_{i=1}^{N} \left[ \frac{1}{T-t+1} \sum_{k=0}^{T-t} I(\overline{x}_i; \mathbf{c}_{T-k,i}|\mathbf{s}) \right] + \varepsilon_{train} \quad (9)$$

*This shows that our marginalization-based sampling procedure effectively governs the same sampling error bound at decoding time by using the averaged mutual information over the decoding trajectory.*

Detailed proofs are provided in B.2. Prop. 3.3 uncovers two advantages of the approximated target distribution over contexts for sampling procedures as follows.

(i) **Our method provides theoretical optimality guarantees.** Benefiting from the implicit government of MI as in Eq. (7), we can approximately maintain an equivalent sampling error bound structure with the ideal one as in Eq. (9). This transforms an inaccessible theoretical optimum into a practically computable objective, ensuring provably controlled sampling error. Empirically, as illustrated in Fig. 1, while different benchmarks exhibit diverse modes of the

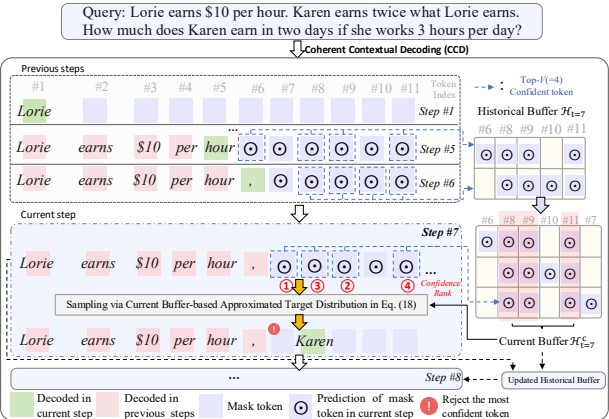

*Figure 2.* Framework of our proposed method. We define a historical buffer $\mathcal{H}_t$ at iteration $t$ that stores the predictive distributions from the most recent $d$ iterations (except for the current iteration $t$) with only the top-$V$ most confident tokens at each iteration. At the current iteration $t$, we also identify mask token positions that appear both in the current top-$V$ set and in the historical buffer to obtain the current buffer $\mathcal{H}_t^c$, which ensures consistently confident tokens with maximized effective contexts to conduct an approximated target distribution-based sampling procedure in Eq. (12). Please zoom in for a better view.

sampling difficulty trajectory, CCD achieves a more converged or lower potential sampling error compared with single-step confidence.

(ii) **An implicit difficulty-aware scheduling mechanism.** As our marginalization implicitly encodes the MI between each token and its evolving contexts, tokens with low MI (low sampling difficulty) are confidently decoded early, while tokens with high MI (high sampling difficulty) are strategically deferred. From a perspective of the overall trajectory, the more contexts allow these difficult tokens to be decoded more easily (at a later stage). Please see the example in Appendix Table 10.

### 3.3. Coherent Contextual Decoding

After the theoretical analysis in Section 3.2, we further clarify that directly implementing the proposed approximated target distribution in Section 3.1 presents two practical challenges: (i) **Non-uniform information quality across diffusion steps.** Commonly, the predictive distribution may not be very informative in the early diffusion steps due to the limited number of contexts, especially for those semantically ambiguous tokens. These early-stage predictions may be regarded as noisy sampling results in MC estimation. (ii) **Prohibitive memory requirements for storing predictive distributions.** Implementing Eq. (5) naively requires storing up to $(T - t) \times N \times |\overline{\mathbb{X}}|$ probability values, *i.e.,* the full distribution for every masked token at every iteration. For long-sequence generation where $N$ is large, this memory overhead becomes prohibitive.

To address these practical challenges, as illustrated in Fig. 2, we introduce a sliding-window historical buffer that maintains only the *most recent and informative predictive distributions*. Specifically, we define a historical buffer $\mathcal{H}_t$ at iteration $t$ that stores the predictive distributions from the most recent $d$ iterations (except for the current iteration $t$) with only the top-$V$ most confident tokens at each iteration. Formally, $\mathcal{H}_t$ can be represented as follows

$$\mathcal{H}_t = \{(i, j, \hat{p}_\theta) : i \in \mathcal{I}_t, j \in \{1, ..., \min(d, T - t)\}\}$$

$$\mathcal{I}_t = \bigcap_{j=1}^{\min(d, T-t)} \{i : i \in \arg\max_{\mathcal{S} \subset \mathcal{K}_{t+j}, |\mathcal{S}| = V} \sum_{i \in \mathcal{S}} -H(\hat{p}_\theta)\} \quad (10)$$

where $\hat{p}_\theta = \hat{p}_\theta(x_i | \mathbf{c}_{t+j,i}, \mathbf{s})$, $\mathcal{I}_t$ identifies mask token positions that appear in the top-$V$ confident sets across all recent $d$ iterations (intersection of position indices), which ensures maximized effective context observations in the later computation of approximated target distribution. Based on $\mathcal{I}_t$, $\mathcal{H}_t$ maintains these selected token position indices, iteration indices, and corresponding predictive distribution in specific iteration index. $\mathcal{H}_t$ can be continually updated with iterations, and it always track consistently confident tokens in the most recent $d$ iterations in a sliding-window manner.

At the current iteration $t$, we also identifies mask token positions that appear both in the current top-$V$ set and in the historical buffer. Finally, we can obtain consistently confident tokens with maximized effective context observations. Formally, the position set $\mathcal{I}_t^c$ of these tokens can be formulated, *i.e.,* $\mathcal{I}_t^c =$

$$\{i : i \in \arg\max_{\mathcal{S} \subset \mathcal{K}_t, |\mathcal{S}| = V} \sum_{i \in \mathcal{S}} -H(\hat{p}_\theta)\} \cap \{i : (i, \cdot, \cdot) \in \mathcal{H}_t\}$$

$$(11)$$

where $\hat{p} = \hat{p}_\theta(x_i | \mathbf{c}_{t+j,i}, \mathbf{s})$. According to $\mathcal{I}_t^c$, we can obtain a current buffer $\mathcal{H}_t^c = \{(i, j, \hat{p}_\theta(x_i | \mathbf{c}_{t+j,i}, \mathbf{s})) : i \in \mathcal{I}_t^c, j \in \{0, ..., \min(d, T - t)\}\}$, which incorporates the history buffer and current predictions, to compute the approximated target distribution $\overline{p}(x_i | \mathbf{s})$ as in Eq. (5) (where the number of MC sampling is $d + 1$ for each selected token). The sampling procedure based on Eq. (1) becomes,

$$p_{t,i} = \overline{p}(x_i \mid \mathbf{s}), \quad \mathcal{J}_t = \arg\max_{\mathcal{S} \subset \mathcal{I}_t^c, |\mathcal{S}| = b_t} \sum_{i \in \mathcal{S}} -H(p_{t,i}) \quad (12)$$

**Discussions.** First, the sliding-window buffer reduces memory complexity from $O((T - t) \times N \times |\mathbb{X}|)$ to $O(d \times V \times |\mathbb{X}|)$, where $d \ll T - t$ and $V \ll N$. Second, by selecting only the top-$V$ most confident predictions (lowest entropy) at each iteration, the buffer automatically filters out noisy, uninformative predictions from early steps.

**Adaptive Sampling Budget.** From Eq. (12), we can find two interesting properties as follows: (i) The selected tokens in $\mathcal{I}_t^c$ are *consistently confident tokens*, which may be more

reliable to be candidates for decoding compared with those tokens selected from single-step confidence in Eq. (1). (ii) The number of these selected tokens, *i.e.*, $|\mathcal{I}_t^c|$ will dynamically vary from 0 to $V$, according to the context sensitivity of generative contents in a diffusion period. For example, $|\mathcal{I}_t^c|$ tends to be large during context-insensitive generation (*e.g.*, inherent templates or formulaic phrases), while it becomes smaller for context-sensitive content requiring careful semantic disambiguation.

These observations is orthogonal to existing sampling procedures that employ a uniform sampling budget $b_t$ at each diffusion step. This raises an intriguing possibility: *can we leverage this natural variation to extend the existing uniform sampling budget scheme to a context-dynamically varied one?* If we do so, there may be two merits. First, semantically ambiguous tokens (high context sensitivity) automatically receive smaller sampling budgets, allowing more iterations to establish proper contexts before decoding. Second, context-insensitive regions can be decoded more aggressively with larger budgets, reducing the overall number of diffusion steps required, which potentially accelerates the diffusion process. The second point is highly related to the generation of the end-of-sentence (EOS) that degrades the efficiency of the decoding process. Please see the empirical example in Appendix Fig. 3 and analyses.

Formally, a context-dynamically varied sampling budget scheme can be represented based on Eq. (12), where the decoding set $\mathcal{J}_t$ is determined adaptively, *i.e.*, Let $\mathcal{S}^* = \arg\max_{\mathcal{S} \subset \mathcal{I}_t^c, |\mathcal{S}|=b_t} \sum_{i \in \mathcal{S}} -H(p_{t,i})$ and $\mathcal{E} = \{i \in \mathcal{I}_t^c : H(p_{t,i}) < \epsilon\}$. The decoding set is determined as:

$$\mathcal{J}_t = \begin{cases} \mathcal{I}_t^c, & \text{if } |\mathcal{I}_t^c| \leq b_t \\ \mathcal{S}^* \cup \mathcal{E}, & \text{if } |\mathcal{I}_t^c| > b_t \end{cases} \tag{13}$$

When $|\mathcal{I}_t^c| > b_t$, we decode the top-$b_t$ tokens plus any additional tokens (ranked from $b_t + 1$ to $|\mathcal{I}_t^c|$) whose entropy over marginalized contexts falls below a small threshold $\epsilon$. This ensures that all selected tokens exhibit both high confidence and predictive consistency across diffusion steps. When $|\mathcal{I}_t^c| \leq b_t$, we decode all $|\mathcal{I}_t^c|$ tokens directly.

## 4. Experiments

**Base Models.** We evaluate our approach using two prominent families of Diffusion Language Models (DLMs): LLaDA (Nie et al., 2025) and Dream (Ye et al., 2025). Specifically, we utilize the `LLaDA-8B-Instruct` and `Dream-7B-Instruct` checkpoints. To ensure reproducibility, we employ their default hyperparameters, official open-source inference codebases, and pretrained weights.

**Baselines.** Our primary baseline is the standard sampling procedure with a uniform budget, as adopted by the base models. Specifically, the Dream series utilizes *negative*

*entropy* to select tokens, while the LLaDA series utilizes the *maximum probability* of single-step predictive distributions. Additionally, we consider two recent decoding strategies, including *top probability margin* (Kim et al., 2025) (namely Margin Conf. hereafter) and *credit decoding* (Ma et al., 2025b) (namely Credit hereafter), where the former works well on small-scale DLMs and the latter achieves gains on LLaDA series. We reproduce these baselines using their default configurations without modification.

**Datasets.** We conduct evaluations across five standard benchmarks covering three distinct capabilities: Mathematical Reasoning: GSM8K (Cobbe et al., 2021) and MATH (Hendrycks et al., 2021). Code Generation: HumanEval (Chen et al., 2021) and MBPP (Austin et al., 2021). Planning: The trip planning benchmark (Zheng et al., 2024a) contains 1600 examples with different difficulties.

**Evaluation.** All tasks are evaluated in a zero-shot setting, with the exception of the Trip benchmark. We utilize the standardized `lm-eval` harness (Gao et al., 2024) to compute accuracy. Beyond accuracy, we report the average number of decoding steps required. As DLM inference is typically memory-bound, the reduction in decoding steps serves as a direct proxy for inference latency speedup.

### 4.1. Implementation Details

**CCD and its Hyperparameters.** Note that CCD is designed to be *plug-and-play, compatible with existing sampling procedures*. Specifically, we can leverage the base model's uncertainty metric (negative entropy for Dream; max probability for LLaDA) to identify the top-$V$ most confident tokens at each iteration. These tokens construct both the current buffer and the sliding-window historical buffer. Note that we also utilize the **block-wise** decoding scheme for the LLaDA series on GSM8K and MATH benchmarks to demonstrate the commonality of CCD.

CCD has two key hyperparameters: the number of confident tokens retained per iteration ($V$) and the history length ($d$). Unless otherwise stated, we set $V = 4$ and $d = 3$ across base models and benchmarks. We provide the specific settings of these hyperparameters in the Appendix Sec. C and sensitivity analyses in the Appendix Sec. A.

**CCD-DS.** The threshold $\epsilon$ controls the sensitivity to marginal entropy, determining when to accelerate generation. Tuning a fixed $\epsilon$ across diverse benchmarks is nontrivial. To address this, we introduce a heuristic based on prediction stability to avoid explicit tuning of $\epsilon$. Specifically, at position $i$, if the token index with the maximum probability remains consistent across all $d+1$ iterations, we infer that the marginal entropy is sufficiently low (implicitly assuming $H(p_{t,i}) < \epsilon$). This stability criterion allows for adaptive acceleration without manual threshold specification.

*Table 1.* Performance of LLaDA 8B and Dream 7B with our proposed method on 5 benchmarks. Note that the diffusion steps are varied across benchmarks and base models, as these settings typically correspond to the best performance on a specific benchmark. Therefore, we follow the base models' default settings without tuning. The subscript values indicate the performance difference compared to the default baseline, where green denotes improvement and red denotes degradation.

| Task | Method | Inference Efficiency | | Performance | Method | Inference Efficiency | | Performance |
|---|---|---|---|---|---|---|---|---|
| | | Diffusion steps↓ | Gains↑ | Score↑(%) | | Diffusion steps↓ | Gains↑ | Score↑(%) |
| **Mathematics Reasoning** | | | | | | | | |
| GSM8K | LLaDA (default) | 512 | $1.00\times$ | 78.08 | Dream (default) | 256 | $1.00\times$ | 81.01 |
| | + Margin Conf. | 512 | $1.00\times$ | $77.17_{-0.91}$ | + Margin Conf. | 256 | $1.00\times$ | $81.72_{+0.71}$ |
| | + Credit | 512 | $1.00\times$ | $78.61_{+0.53}$ | + Credit | 256 | $1.00\times$ | $81.88_{+0.87}$ |
| | + CCD | 512 | $1.00\times$ | $\mathbf{78.90}_{+0.82}$ | + CCD | 256 | $1.00\times$ | $\mathbf{82.51}_{+1.50}$ |
| | + CCD-DS | $463_{-49.0}$ | $1.11\times_{+0.11}$ | $78.24_{+0.16}$ | + CCD-DS | $145.2_{-110.8}$ | $1.76\times_{+0.76}$ | $\underline{82.01}_{+1.00}$ |
| Math | LLaDA (default) | 512 | $1.00\times$ | 37.00 | Dream (default) | 512 | $1.00\times$ | 40.90 |
| | + Margin Conf. | 512 | $1.00\times$ | $36.50_{-0.50}$ | + Margin Conf. | 512 | $1.00\times$ | 40.90 |
| | + Credit | 512 | $1.00\times$ | $37.30_{+0.30}$ | + Credit | 512 | $1.00\times$ | $41.10_{+0.20}$ |
| | + CCD | 512 | $1.00\times$ | $\mathbf{37.40}_{+0.40}$ | + CCD | 512 | $1.00\times$ | $\mathbf{41.24}_{+0.34}$ |
| | + CCD-DS | $379.2_{-132.8}$ | $1.35\times_{+0.35}$ | $37.20_{+0.20}$ | + CCD-DS | $323.5_{-188.5}$ | $1.58\times_{+0.58}$ | $41.20_{+0.30}$ |
| **Code Generation** | | | | | | | | |
| HumanEval | LLaDA (default) | 512 | $1.00\times$ | 36.50 | Dream (default) | 768 | $1.00\times$ | 52.66 |
| | + Margin Conf. | 512 | $1.00\times$ | $35.30_{-1.20}$ | + Margin Conf. | 768 | $1.00\times$ | $53.65_{+0.99}$ |
| | + Credit | 512 | $1.00\times$ | $35.90_{-0.60}$ | + Credit | 768 | $1.00\times$ | $55.71_{+3.05}$ |
| | + CCD | 512 | $1.00\times$ | $\mathbf{40.90}_{+4.40}$ | + CCD | 768 | $1.00\times$ | $\mathbf{56.71}_{+4.05}$ |
| | + CCD-DS | $332.0_{-180.0}$ | $1.54\times_{+0.54}$ | $40.80_{+4.30}$ | + CCD-DS | $249.8_{-518.2}$ | $3.08\times_{+2.08}$ | $56.61_{+3.95}$ |
| MBPP | LLaDA (default) | 256 | $1.00\times$ | 39.20 | Dream (default) | 1024 | $1.00\times$ | 58.00 |
| | + Margin Conf. | 256 | $1.00\times$ | $38.80_{-0.40}$ | + Margin Conf. | 1024 | $1.00\times$ | 58.00 |
| | + Credit | 256 | $1.00\times$ | $\mathbf{40.60}_{+1.40}$ | + Credit | 1024 | $1.00\times$ | $58.40_{+0.40}$ |
| | + CCD | 256 | $1.00\times$ | 39.20 | + CCD | 1024 | $1.00\times$ | $\mathbf{59.00}_{+1.00}$ |
| | + CCD-DS | $220.30_{-35.7}$ | $1.16\times_{+0.16}$ | 39.20 | + CCD-DS | $277.50_{-746.5}$ | $3.69\times_{+2.69}$ | 58.00 |
| **Planning** | | | | | | | | |
| Trip Plan | LLaDA (default) | 256 | $1.00\times$ | 10.40 | Dream (default) | 256 | $1.00\times$ | 15.10 |
| | + Margin Conf. | 256 | $1.00\times$ | $\underline{10.80}_{+0.40}$ | + Margin Conf. | 256 | $1.00\times$ | $15.90_{+0.80}$ |
| | + Credit | 256 | $1.00\times$ | $10.61_{+0.21}$ | + Credit | 256 | $1.00\times$ | $15.60_{+0.50}$ |
| | + CCD | 256 | $1.00\times$ | $\underline{10.80}_{+0.40}$ | + CCD | 256 | $1.00\times$ | $\underline{16.93}_{+1.83}$ |
| | + CCD-DS | $112.5_{-143.5}$ | $2.27\times_{+1.27}$ | $\mathbf{11.50}_{+1.10}$ | + CCD-DS | $75.20_{-180.80}$ | $3.40\times_{+2.40}$ | $\mathbf{19.01}_{+3.91}$ |

## 4.2. Main results

The comparison results can be found in the Table 1. One can have two observations as follows.

**- Consistent Performance Improvement with CCD.** We begin by evaluating the effectiveness of our proposed CCD method under a uniform sampling budget of $b_t = 1$. As shown in Table 1, the deployment of CCD consistently enhances performance metrics across all five benchmarks for both the LLaDA-8B-Instruct and Dream-7B-Instruct models. These gains are particularly pronounced in complex reasoning tasks; for example, the Dream model achieves a score increase of $+4.05$ on HumanEval and $+1.83$ on the Trip Plan benchmark. This validates that our approximated target distribution effectively guides the model toward higher-quality outputs. Notably, CCD also demonstrates robust compatibility with the block-wise decoding scheme of the LLaDA series on the GSM8K and MATH benchmarks.

**- Diverse Efficiency Enhancement across Base Models and Benchmarks with CCD-DS.** Upon applying the adaptive CCD-DS strategy, we have three significant observations as follows. (i) *Efficiency gains rely on the capacity of*

*base models.* CCD-DS has more obvious efficiency gains on the Dream, which may be reasonable as the Dream with strong base performance is prone to more consistently confident tokens compared with LLaDA, leading to a large $|\mathcal{I}_t^c|$ in Eq. (13). (ii) *Efficiency gains rely on the generative characteristic of benchmarks.* Compared with open-end generative processes on mathematics and coding tasks, the planning task on the Trip benchmark is more obvious to be accelerated using CCD-DS, which is reasonable as a large proportion of generative tokens exist in the prompt, enhancing the generative potential of consistently confident tokens. (iii) *CCD-DS does not degrade generation quality.* Even, CCD-DS outperforms the fixed-step baseline CCD on the Trip Plan benchmark. On the Trip, the Dream with CCD-DS achieves a score of $19.01\%$ vs. $16.93\%$ with CCD.

## 4.3. Comparison on Highly Parallel Decoding

Realizing the high parallelism in existing DLMs remains challenging. We explore the performance of different decoding methods on two scenarios.

**- Fixed high decoding budgets.** As shown in Table 2, we

*Table 2.* Highly parallel decoding performance of Dream 7B using different sampling approaches.

| Task | Method | $b_t$=1 | $b_t$=2 | $b_t$=3 | Avg.(%) | Task | Method | $b_t$=1 | $b_t$=2 | $b_t$=3 | Avg.(%) |
|---|---|---|---|---|---|---|---|---|---|---|---|
| GSM8K | Dream default | 81.01 | 78.47 | 69.07 | 76.18 | HumanEval | Dream default | 52.66 | 48.17 | 23.17 | 41.33 |
| | + Margin Conf. | 81.72 | 78.47 | 68.58 | 76.26 | | + Margin Conf. | 53.65 | 48.17 | 23.17 | 41.66 |
| | + Credit | 81.88 | 76.48 | 67.78 | 75.38 | | + Credit | 55.71 | 40.24 | 20.73 | 38.89 |
| | + CCD | 82.51 | 79.00 | 78.55 | **80.02** | | + CCD | 56.71 | 49.41 | 45.73 | **50.62** |
| MATH | Dream default | 40.90 | 32.89 | 22.87 | 32.22 | MBPP | Dream default | 58.00 | 43.80 | 29.20 | 43.67 |
| | + Margin Conf. | 40.90 | 33.92 | 23.52 | 32.78 | | + Margin Conf. | 58.00 | 43.80 | 28.30 | 43.37 |
| | + Credit | 41.10 | 33.28 | 19.96 | 31.45 | | + Credit | 58.40 | 40.20 | 27.60 | 41.93 |
| | + CCD | 41.24 | 35.68 | 30.12 | **35.67** | | + CCD | 59.00 | 50.60 | 43.60 | **51.06** |

*Table 3.* Performance comparison of LLaDA 8B with Fast-dLLM and CCD-DS method on 4 benchmarks.

| Task | Metric | LLaDA (default) | +Fast-dLLM | +CCD-DS | Task | Metric | LLaDA (default) | +Fast-dLLM | +CCD-DS |
|---|---|---|---|---|---|---|---|---|---|
| GSM8K | Performance↑(%) | 78.07 | 78.01$_{-0.06}$ | **78.77**$_{+0.70}$ | MATH | Performance↑(%) | 37.00 | 35.80$_{-1.20}$ | **36.30**$_{-0.70}$ |
| | Efficiency↑ | 1.00× | 4.80× | 4.79× | | Efficiency↑ | 1.00× | 3.72× | 3.68× |
| HumanEval | Performance↑(%) | 36.50 | 35.90$_{-0.60}$ | **38.00**$_{+1.50}$ | MBPP | Performance↑(%) | 39.02 | **38.80**$_{-0.22}$ | 38.60$_{-0.42}$ |
| | Efficiency↑ | 1.00× | 4.13× | 4.02× | | Efficiency↑ | 1.00× | 2.85× | 2.79× |

*Table 4.* Comparison of reasoning processes and diffusion trajectories on a case of GSM8K. See Table 8 for full question and responses.

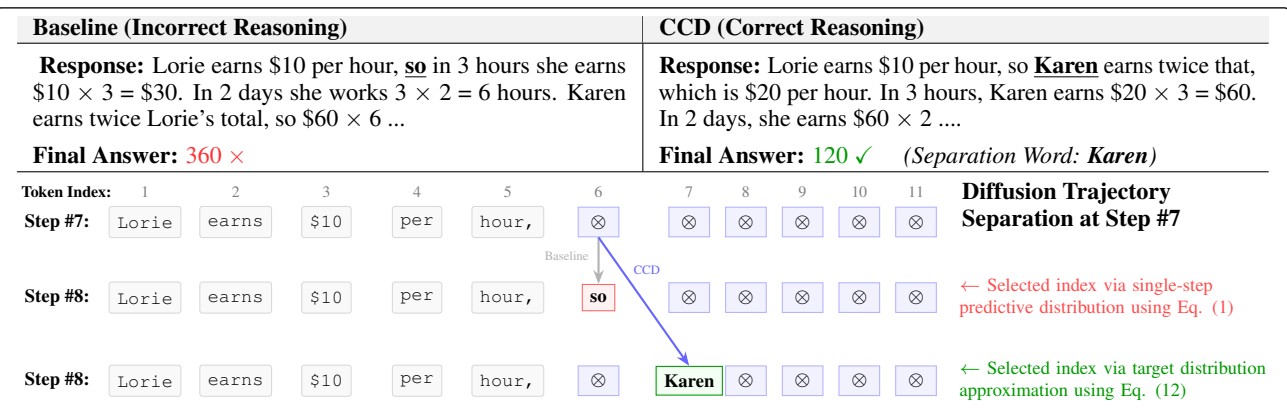

increase the decoding budget for each step. If the number of $|\mathcal{I}_t^c|$ in Eq. (11) for CCD is lower than the fixed decoding budget, we will directly make up for the deficiencies based on single-step confidence priority. As we can see, CCD significantly outperforms existing decoding approaches. This is reasonable because CCD naturally prioritizes those tokens that exhibit high *confidence* and *predictive consistency* across steps, leading to a more controllable sampling error rate (as revealed in Prop. 3.3) in highly parallel conditions. **- Dynamic decoding budgets.** As illustrated in Table 3, we compare CCD-DS with confidence threshold-based dynamic method, *i.e.,* Fast-dLLM (Wu et al., 2025). To match Fast-dLLM's efficiency performance, CCD-DS increases the size of $V$ for each task (where GSM8K and Math are 8; HumanEVAL and MBPP are 10), leading to a slight performance difference with Table 1. As we can see, CCD-DS achieves milder performance degradation in most cases at a similar level of efficiency gain.

**Case study.** As shown in Table 4, CCD enables rejection of the most confident token at the current step, leading to accurate final results. Specifically, although the top-1 confident token locates 6-th token position by computing the negative entropy of single-step predictive distirbution, CCD accepts a more context-insensitive token ("Karan" rather than "so"), avoiding a potential suboptimal decision. More cases in the Appendix Sec. D.

### 4.4. More Analyses

**Effect of historical iteration step parameter.** We conduct an ablation over the historical iteration step parameter $d$, while fixing the buffer size as $V = 4$ on Dream model. As illustrated in Table 5, The results indicate that $d = 3$ provides the best average performance, achieving a strong trade-off: it benefits from longer historical consistency while avoiding the diminishing returns (and potential noise accumulation) observed when d grows further.

**Component ablation.** We conducte the ablation to analyze the different variants of CCD. *Average only:* This variant simply computes the margin entropy of d steps for all mask tokens, which does not toke the single-step confidence into account. *Averaging with top-V:* This variant computes the

*Table 5.* Ablation over the historical iteration step parameter $d$, with the buffer size fixed at $V = 4$. When $d = 0$, the overall method degrades to single-step decoding.

| $d$ | GSM8K | Math | HumanEval | MBPP | Avg |
|---|---|---|---|---|---|
| 0 | 81.01 | 40.90 | 52.66 | 58.00 | 58.00 |
| 1 | 81.65 | 41.20 | 55.32 | 58.20 | 58.20 |
| 2 | 82.26 | 40.46 | 57.92 | 58.40 | 58.40 |
| 3 | **82.51** | **41.24** | 56.71 | **59.00** | **59.00** |
| 4 | 81.90 | 40.54 | **57.99** | 56.80 | 56.80 |

*Table 6.* Ablation on the design choices of CCD on GSM8K.

| Variant | GSM8K |
|---|---|
| Default | 81.01 |
| + Averaging only | 81.90 |
| + Averaging with top-$V$ | 80.39 |
| + Averaging with top-$V$ and intersection | **82.51** |

margin entropy of top-V mask tokens of each step. The number of computed tokens ranges from V (the top-V tokens are equal across d steps) to 3V (the top-V tokens are different across d steps). Correspondingly, the number of the prediction of the computed tokens rangers from d to 1. *Averaging with top-V and intersection:* This variant is equal to the CCD's implementation. As illustrated in Table 6, the averaging only-based variant and the CCD can improve the accuracy compared to the the default single-step decoding algorithm, but the variant based on averaging with top-$V$ degrades the performance. This phenomenon is reasonable because 1) Compared to CCD and the averaging only-based variant, the variant based on averaging with top-$V$ suffers from unstable computation of marginal entropy. For example, a token, which is only included in the top-V confident token of the $(d$-1)th step, still is considered for computing marginal entropy. Obviously, such a token has a limited correlation to the current context for decoding, leading to compromised performance. 2) Compared to the CCD, the averaging only-based variant will consider all tokens' predictions in each step, which may naturally incur the noisy and uninformative predictions. Instead, the intersection across steps using by CCD can be regarded as a more strict selection metric, leading to a more reliable and computationally-efficient marginal entropy realization.

**Limitations.** First, due to the additional computation in a post-processing manner, CCD will slightly increase the runtime of each forward process. However, it may be also a trade-off between efficiency and effectiveness. Second, the sliding-window top-$V$ intersection introduces three approximations (including the window truncation, the Top-$V$ filtering, and the intersection) relative to the full trajectory average. It desires to explore more theoretically-grounding solutions in the future.

# 5. Related works

Compared to diffusion processes in continuous language domains (Campbell et al., 2022), the scale of discrete diffusion (Lou et al., 2023; Austin et al., 2021; Shi et al., 2024; Zheng et al., 2024b) models has expanded significantly, recently reaching the 7–8 billion parameter regime, *e.g.*, LLaDA (Nie et al., 2025) and Dream (Ye et al., 2025) with competitive performance. Prominent commercial implementations (Song et al., 2025b; DeepMind, 2025; Labs et al., 2025) further illustrate the maturity of this field. Alongside architectural advancements, researchers are actively optimizing the inference phase to achieve a better balance of latency and performance. This direction largely centers on two streams: implementing efficient caching protocols (*e.g.* caching techniques (Wu et al., 2025; Ma et al., 2025a; Liu et al., 2025; Song et al., 2025a)) and developing efficient and effective decoding algorithms (Yu et al., 2025; Li et al., 2025b; Hong et al., 2025; Yang et al., 2025; Li et al., 2025a). For these decoding approaches, there are concurrent works that also consider historical decoding steps. Wang et al. (2025) introduced temporal self-consistency voting, which selects the most reliable and consistent trajectory in a postprocessing manner. dInfer (Ma et al., 2025b) proposes credit decoding, where the predictive distribution is updated by continually fusing logit and credit scores.

In contrast, CCD implicitly models the consistency of historical steps using the *conditional mutual information*, which can naturally prioritize those tokens that exhibit high *confidence* and *predictive consistency* across steps without postprocessing and long-term prediction accumulation.

**Difference with dInfer's credit decoding.** First, one of the most differences is that CCD derives its selection criterion from conditional mutual information (Prop. 3.1), connecting to the sampling error bound (Prop. 3.3) for better performance gains. Instead, dInfer lacks this theoretical framework. Second, dInfer needs to continuously fuse logits via in a long-term credit scores, which may limite its impact on long-context scenarios with the potential over-smooth risk. Instead, CCD uses intersection of top-$V$ confident sets across d recent steps. This intersection-based approach explicitly targets predictive consistency to prioritize those tokens with high confidence and predictive consistency.

# 6. Conclusion

We presented CCD, a new framework of DLM sampling by leveraging historical contexts and grounding in conditional mutual information to reject sub-optimal paths without compromising coherence. Notably, CCD achieves performance gains under a uniform sampling scheme and milder performance degradation under highly parallel decoding scenarios. As a variant, CCD-DS shows a better performance and efficiency balance on dynamic decoding scenarios.

## Acknowledgment

This work was supported in part by the Hong Kong Innovation and Technology Commission (ITC) (InnoHK Project CIMDA), in part by the Institute of Digital Medicine of City University of Hong Kong (CityU) under Project 9229503, in part by Chow Sang Sang Donation and Matching Fund (Project 9229161), and in part by the CityU under Project 7006087.

## Impact Statement

This paper presents work whose goal is to advance the field of Machine Learning. There are many potential societal consequences of our work, none which we feel must be specifically highlighted here.

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

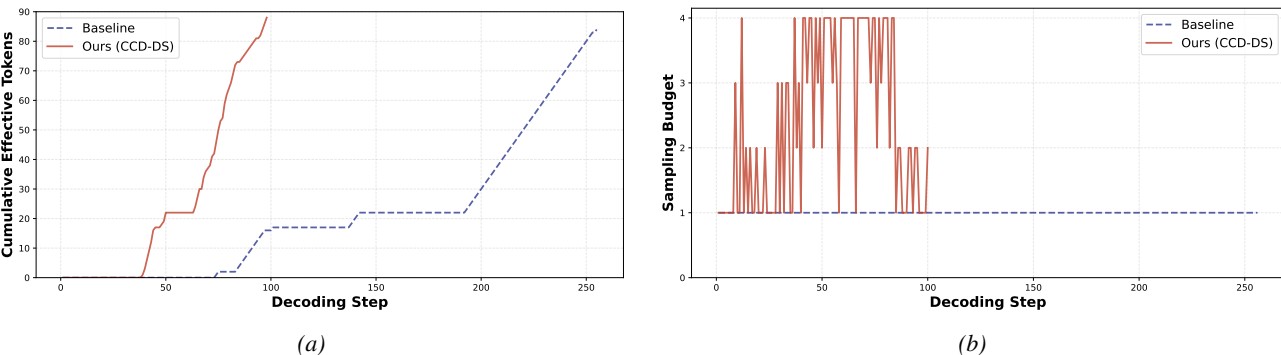

*(a)* *(b)*

*Figure 3.* This analysis investigates the Cumulative effective tokens (CET) and sampling budget using the Dream model on the Trip benchmark. The sequence length is set to 256. (a) We calculate the CET by excluding padding/EOS tokens from the total generated tokens. The decoding process includes periods where multiple EOS tokens are generated (visible as **plateaus**). (b) Sampling budget of each diffusion step of different sampling procedures. For our method, the decoding process is early stopped because all mask tokens are decoded.

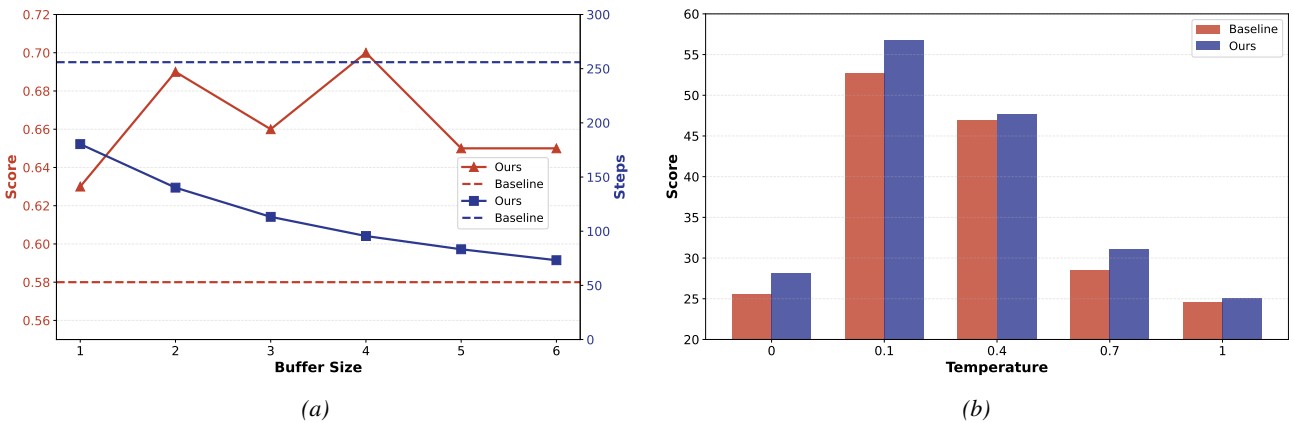

*(a)* *(b)*

*Figure 4.* Hyperparameter Analysis on buffer size and temperature coefficients using the Dream model. (a) The trade-off between score and computational steps as buffer size varies on the subset (City=3) of Trip benchmark. (b) Performance comparison across different temperature coefficients on the HumanEval benchmark.

# A. More analyses

**Decoding Efficiency Bottleneck: Generation Plateau Period of DLMs.** As illustrated in Figure 3(a), instead of discretely generating EOS tokens across the decoding process, we observe that the decoding process of DLMs actually suffers from continual generation periods of EOS tokens, *i.e.,* multiple plateau periods in the increment of effective tokens (excluding EOS tokens). Since we cannot know the ground-truth number of generative responses beforehand, a larger number of mask tokens is typically adopted, resulting in the inevitable generation of these EOS tokens. For the existing uniform sampling budget scheme, it is obvious that the decoding efficiency is low during these plateau periods.

In contrast, as illustrated in Figure 3(b), our proposed method can adaptively obtain an ad-hoc sampling budget across the decoding process with maximum budgets set to 4. Therefore, our proposed method can pass through the plateau periods more efficiently by decoding more EOS tokens in each step. Meanwhile, we observe that our proposed method can automatically obtain smaller sampling budgets (1 or 2) at the late stage of the decoding process, which ensures the accuracy of generation, as this stage usually includes semantically ambiguous tokens.

**Impact of Buffer Size.** Our proposed method relies on the size of the historical buffer for balancing efficiency and effectiveness gains. As shown in Figure 4a, our method consistently outperforms the baseline across all buffer sizes in terms of both performance and inference speed. The accuracy peaks at buffer size 4 (70%), representing a 20.7% improvement over the baseline (58%). Simultaneously, our approach requires substantially fewer steps across all configurations, with the reduction ranging from 29.5% (180.37 vs. 256 steps at buffer size 1) to 71.4% (73.21 vs. 256 steps at buffer size 6).

---

**Algorithm 1** Coherent Confident Decoding (CCD)

---

**Require:** Model $p_\theta$, prompt $s$, sequence length $N$, steps $T$, window size $d$, confidence filter $V$, budget schedule $\{b_t\}_{t=T}^1$
**Ensure:** Generated sequence $x$

1: $x \leftarrow [\text{MASK}]^N$ {Initialize all positions as masked}
2: $\mathcal{B} \leftarrow []$ {Historical buffer}
3: **for** $t = T$ **down to** 1 **do**
4:     $\mathcal{K}_t \leftarrow \{i : x_i = [\text{MASK}]\}$ {Current masked positions}
5:     $b \leftarrow \min(b_t, |\mathcal{K}_t|)$ {Effective budget}
6:     *// —- Forward pass —-*
7:     **for all** $i \in \mathcal{K}_t$ **do**
8:         $\hat{p}_i \leftarrow \hat{p}_\theta(x_i \mid c_{t,i}, s)$ {Predictive distribution}
9:         $h_i \leftarrow H(\hat{p}_i)$ {Entropy}
10:     **end for**
11:     *// —- Step 1: Current top-V confident positions —-*
12:     $V' \leftarrow \min(V, |\mathcal{K}_t|)$
13:     $\text{TopV}_t \leftarrow V'$ positions in $\mathcal{K}_t$ with smallest $h_i$
14:     *// —- Step 2: Coherent candidate set (Eq. 8 & 9) —-*
15:     **if** $|\mathcal{B}| \geq 1$ **then**
16:         $\mathcal{I}_t \leftarrow \bigcap_{(\mathcal{P}_j, \cdot) \in \mathcal{B}} \mathcal{P}_j$ {Intersection of historical top-$V$ sets}
17:         $\mathcal{I}_t^c \leftarrow \text{TopV}_t \cap \mathcal{I}_t$
18:     **else**
19:         $\mathcal{I}_t^c \leftarrow \emptyset$
20:     **end if**
21:     *// —- Step 3: Marginalized distributions for coherent candidates —-*
22:     **for each** $i \in \mathcal{I}_t^c$ **do**
23:         $\mathcal{D}_i \leftarrow \{\hat{p}_i\} \cup \{$stored dist. for position $i$ from each entry in $\mathcal{B}\}$
24:         $\bar{p}_i \leftarrow \dfrac{1}{|\mathcal{D}_i|} \sum_{p \in \mathcal{D}_i} p$ {MC average (Eq. 5)}
25:         $\bar{h}_i \leftarrow H(\bar{p}_i)$
26:     **end for**
27:     *// —- Step 4: Determine decoding set $\mathcal{J}_t$ —-*
28:     **if** $|\mathcal{I}_t^c| \geq b$ **then**
29:         $\mathcal{J}_t \leftarrow b$ positions in $\mathcal{I}_t^c$ with smallest $\bar{h}_i$ {Eq. 10}
30:     **else if** $|\mathcal{I}_t^c| > 0$ **then**
31:         $\mathcal{J}_t \leftarrow \mathcal{I}_t^c$ {Decode all coherent candidates}
32:         $\mathcal{R} \leftarrow (b - |\mathcal{I}_t^c|)$ positions in $(\mathcal{K}_t \setminus \mathcal{I}_t^c)$ with smallest $h_i$
33:         $\mathcal{J}_t \leftarrow \mathcal{J}_t \cup \mathcal{R}$ {Supplement with single-step confidence}
34:     **else**
35:         $\mathcal{J}_t \leftarrow b$ positions in $\mathcal{K}_t$ with smallest $h_i$ {Fallback}
36:     **end if**
37:     *// —- Step 5: Decode selected positions —-*
38:     **for each** $i \in \mathcal{J}_t$ **do**
39:         **if** $i \in \mathcal{I}_t^c$ **then**
40:             $x_i \leftarrow \arg\max \bar{p}_i$ {Marginalized distribution}
41:         **else**
42:             $x_i \leftarrow \arg\max \hat{p}_i$ {Single-step distribution}
43:         **end if**
44:     **end for**
45:     *// —- Step 6: Update historical buffer (sliding window) —-*
46:     $\mathcal{B}.\text{push\_front}\big((\text{TopV}_t, \{(i, \hat{p}_i) : i \in \text{TopV}_t\})\big)$
47:     **if** $|\mathcal{B}| > d$ **then**
48:         $\mathcal{B}.\text{pop\_back}()$ {Remove oldest entry}
49:     **end if**
50: **end for**
51: **return** $x$

---

Notably, buffer size 4 achieves an optimal balance, delivering the highest accuracy while requiring only 95.54 steps—a 62.7% reduction in diffusion step. This demonstrates that our method effectively leverages the historical buffer to accelerate decoding process while significantly enhance performance.

**Robustness to Temperature Coefficients.** As illustrated in Figure 4b, our method demonstrates strong robustness across varying temperature coefficients. Both methods exhibit similar trends, with performance peaking at temperature 0.1 (52.66% for baseline, 56.71% for ours) and declining at extreme values. However, our method consistently outperforms the baseline across the entire temperature range, with improvements of 9.8%, 7.7%, 1.5%, 9.0%, and 2.0% at temperatures 0, 0.1, 0.4, 0.7, and 1.0, respectively. The performance advantage is most pronounced at lower temperatures (0 and 0.1) and near the upper bound (0.7), suggesting our approach is particularly effective when exploration-exploitation balance is critical. Importantly, even at suboptimal temperatures (0.7 and 1.0), our method maintains meaningful gains, demonstrating robustness to hyperparameter choices.

# B. Proofs

## B.1. Proof for Connection via Mutual Information

**Proposition B.1** (Connection via Mutual Information)**.** *The approximated target distribution obtained by marginalizing over contexts is related to the decoding-time single-step predictive distribution through the conditional mutual information between contexts and token predictions:*

$$H(x_i|\mathbf{s}) = H(x_i|\mathbf{c}, \mathbf{s}) + I(x_i; \mathbf{c}|\mathbf{s}) \propto H(x_i|\mathbf{c}_{.,i}, \mathbf{s}) + I(x_i; \mathbf{c}|\mathbf{s}) \tag{14}$$

*where $I(x_i; \mathbf{c}|\mathbf{s})$ is the conditional mutual information between $x_i$ and $\mathbf{c}$ given $\mathbf{s}$, and $\mathbf{c}_{.,i}$ is the specific decoding-time context.*

*Proof.* Since entropy directly measures the uncertainty of distributions and determines the selection of tokens to decode in DLMs (tokens with lower entropy are typically decoded first), we analyze the relationship through entropy decomposition. We start with the chain rule for conditional entropy applied in two different orders:

$$H(x_i, \mathbf{c}|\mathbf{s}) = H(\mathbf{c}|\mathbf{s}) + H(x_i|\mathbf{c}, \mathbf{s}) = H(x_i|\mathbf{s}) + H(\mathbf{c}|x_i, \mathbf{s})$$

Rearranging yields:

$$H(x_i|\mathbf{s}) = H(x_i|\mathbf{c}, \mathbf{s}) + \underbrace{[H(\mathbf{c}|\mathbf{s}) - H(\mathbf{c}|x_i, \mathbf{s})]}_{I(x_i; \mathbf{c}|\mathbf{s})}$$
$$= H(x_i|\mathbf{c}, \mathbf{s}) + I(x_i; \mathbf{c}|\mathbf{s}) \tag{15}$$

For the specific decoding-time context $\mathbf{c}_{.,i}$, we have $H(x_i|\mathbf{c}, \mathbf{s}) \approx H(x_i|\mathbf{c}_{.,i}, \mathbf{s})$ when $\mathbf{c}_{.,i}$ is a sufficiently representative sample from the context distribution. $\square$

Proposition 3.1 uncovers two advantages of our proposed approximated target distribution over contexts for sampling procedures:

- **Additional consideration of statistical dependencies.** When contexts are informative about $x_i$ (high mutual information), our method captures additional statistical dependencies through the $I(x_i; \mathbf{c}|\mathbf{s})$ term, leading to better-calibrated predictions.

- **Flexible connection with the existing sampling procedure.** Our approach exhibits flexibility—when $x_i$ and $\mathbf{c}$ are conditionally independent given $\mathbf{s}$, we have $I(x_i; \mathbf{c}|\mathbf{s}) = 0$, and our method naturally degrades to the existing single-context method with $H(x_i|\mathbf{s}) = H(x_i|\mathbf{c}_{.,i}, \mathbf{s})$. This shows that our marginalization approach is a strict generalization that adapts to the underlying statistical structure of the problem.

### B.2. Proof for Sampling Error Governance via Marginalized Contexts

**Proposition B.2** (Sampling Error Governance via Marginalized Contexts). *Under conditional generation with our approximated target distribution $\overline{p}(x_i|\mathbf{s})$ obtained by marginalizing over contexts, as $t \to 0$, the sampling error bound is governed by*

$$\mathbb{E}\big[\mathsf{KL}(p(\overline{\mathbf{x}}|\mathbf{s}) \parallel p(\hat{\mathbf{x}}|\mathbf{s}))\big] \leq \frac{G}{T} \sum_{i=1}^{N} I(\overline{x}_i; \overline{x}_{-i}|\mathbf{s}) + \varepsilon_{train} \quad \approx \frac{G}{T} \sum_{i=1}^{N} \left[ \frac{1}{T-t+1} \sum_{k=0}^{T-t} I(\overline{x}_i; \mathbf{c}_{T-k,i}|\mathbf{s}) \right] + \varepsilon_{train} \quad (16)$$

*This shows that our marginalization-based sampling procedure effectively governs the same sampling error bound at decoding time by using the averaged mutual information over the decoding trajectory.*

*Proof.* From Proposition 3.1, we established that marginalizing over contexts yields:

$$H(\overline{x}_i|\mathbf{s}) = H(\overline{x}_i|\mathbf{c}, \mathbf{s}) + I(\overline{x}_i; \mathbf{c}|\mathbf{s}) \tag{17}$$

During the DLM decoding process, at each iteration $T - k$ before position $i$ is decoded, we have a context $\mathbf{c}_{T-k,i}$ that progressively refines toward the ground-truth context $\overline{x}-i$, which can be viewed as the limiting case of this refinement process. By marginalizing over the trajectory of contexts:

$$\overline{p}(x_i|\mathbf{s}) = \frac{1}{T-t+1} \sum_{k=0}^{T-t} \hat{p}_\theta(x_i|x_{T-k,i} = \mathbf{M}, \mathbf{c}_{T-k,i}, \mathbf{s}) \tag{18}$$

This marginalization effectively computes an averaged mutual information $\frac{1}{T-t+1} \sum_{k=0}^{T-t} I(\overline{x}_i; \mathbf{c}_{T-k,i}|\mathbf{s})$.

The mutual information $I(\overline{x}_i; \overline{x}_{-i} \mid \mathbf{s})$ represents the dependency of token $i$ on the full context. In our approximation, we substitute the full context $\overline{x}_{-i}$ with the evolving contexts $\{\mathbf{c}_{T-k,i}\}$ from the decoding trajectory. Since $\overline{p}(x_i \mid \mathbf{s})$ averages the predictive distributions, the effective entropy used for selection is derived from this ensemble. As $t \to 0$, the context $\mathbf{c}_{t,i}$ converges toward the full unmasked sequence $\overline{x}_{-i}$. Therefore, the term $\frac{1}{T-t+1} \sum_{k=0}^{T-t} I(\overline{x}_i; \mathbf{c}_{T-k,i} \mid \mathbf{s})$ serves as a tractable lower-bound estimate of the true mutual information complexity, governing the sampling error dynamically during decoding as $t \to 0$:

$$\mathbb{E}\big[\mathsf{KL}(p(\overline{\mathbf{x}}|\mathbf{s}) \parallel p(\hat{\mathbf{x}}|\mathbf{s}))\big] \leq \frac{G}{T} \sum_{i=1}^{N} \left[ \frac{1}{T-t+1} \sum_{k=0}^{T-t} I(\overline{x}_i; \mathbf{c}_{T-k,i}|\mathbf{s}) \right] + \varepsilon_{train} \tag{19}$$

$\square$

Proposition 3.3 uncovers two advantages of our proposed approximated target distribution over contexts for sampling procedures:

- **Our method provides theoretical optimality guarantees.** Benefiting from the implicit introduction of mutual information as in Eq. (14), we can approximately maintain an equivalent sampling error bound structure with the ideal one. This transforms an inaccessible theoretical optimum into a practically computable objective, ensuring provably controlled sampling error at every decoding step.

- **An implicit difficulty-aware scheduling mechanism.** As our marginalization implicitly encodes the mutual information between each token and its evolving contexts, this may create a property: tokens with low mutual information (low sampling difficulty) are confidently decoded early, while tokens with high mutual information (high sampling difficulty) are strategically deferred. From a perspective of the overall trajectory, the more contexts (more unmasked tokens) allow these difficult tokens to be decoded more easily (at a later stage).

## C. Exprimental details

### C.1. Hyperparameter

As shown in Table 7, we provide the full list of used hyperparameters.

*Table 7.* Hyperparameter settings for CCD across different base models and benchmarks. "–" indicates the hyperparameter is not applicable or not used.

| Base Model | Hyperparameter | GSM8K | MATH | MBPP | HumanEval |
|---|---|---|---|---|---|
| Dream-7B-Instruct | Max New Tokens | 256 | 512 | 1024 | 768 |
| | Diffusion Steps | 256 | 512 | 1024 | 768 |
| | History Length $d$ | 3 | 2 | 3 | 3 |
| | Temperature | 0.1 | 0.1 | 0.1 | 0.1 |
| | Top-$p$ | 0.9 | 0.9 | 0.9 | 0.9 |
| | Top-$k$ | – | – | – | – |
| | Buffer Size | 4 | 4 | 4 | 4 |
| LLaDA-8B-Instruct | Max New Tokens | 512 | 512 | 256 | 512 |
| | Diffusion Steps | 512 | 512 | 256 | 512 |
| | Block Length | 8 | 64 | 256 | 512 |
| | History Length $d$ | 3 | 3 | 2 | 3 |
| | Temperature | 0.0 | 0.0 | 0.0 | 0.0 |
| | Top-$p$ | 0.9 | 0.9 | 0.9 | 0.9 |
| | Top-$k$ | – | – | – | – |
| | Buffer Size | 4 | 4 | 4 | 4 |

## D. Case study

**– Our proposed method can encourage more reasonable generative trajectories.** As demonstrated in Table 9, our method leverages global context to make superior decoding decisions at critical divergence points. At Step #25 after `"The first glass costs $5"`, the baseline selects the period `"."` (25th mask token) based on single-step predictive confidence. In contrast, our approximated target distributions identify the 28th mask token "second" as optimal, enabling `", and every second glass costs..."`. This choice leads to a superior reasoning strategy: establishing the general pricing pattern first, then performing a consolidated calculation (`"8 glasses at $5 and 8 glasses at $3"`).

This exemplifies a key mechanistic advantage: while single-step confidence gravitates toward syntactically safe choices (sentence-ending punctuation), our cross-iteration consistency recognizes that continuing with "second" enables more elegant problem decomposition. Trajectory-level optimization fundamentally improves solution quality in reasoning tasks where strategic planning outweighs local fluency.

**– Our proposed method can automatically defer the decoding of semantically ambiguous tokens.** As shown in Table 10, when three mask tokens remain, the baseline has already incorrectly decoded "8" (should be "7") at position "Day 12-1_". This premature decoding occurs because the baseline's uniform sampling budget forces token selection based on a single-step predictive distribution alone. In contrast, our method identifies this token as having high context sensitivity and shows inconsistent predictions across iterations in the historical buffer. Therefore, our proposed method automatically defers its decoding by maintaining it as masked.

This deferral is crucial for arithmetic-dependent reasoning tasks. By keeping these ambiguous tokens masked longer, our method can leverage the constraint that total days must sum to 22, which becomes clearer asthe surrounding context solidifies. The baseline's early mistake cascades through subsequent tokens ("Day 18" instead of "Day 17"), while our sampling procedure naturally preserves tokens with low cross-iteration agreement for later stages when sufficient context is available, preventing error accumulation in structured reasoning where tokens have strong interdependencies.

**Task:** *Lorie earns $10 per hour. Karen earns twice what Lorie earns. How much does Karen earn in two days if she works 3 hours per day?*  `Incorrect`  `Correct`  ⊗ Mask token

**Baseline:** `Lorie earns $10 per hour, so in 3 hours she earns $10 x 3 = $30.`
`Karen earns twice what Lorie earns, so in 3 hours she earns 2 x $30 = $60.`
`Karen works 3 hours per day, so in 2 days she works 3x 2 = 6 hours.`
`Therefore, in 2 days, Karen earns $60 x 6 = $360.` *It should be $60 x 2, rather than $60 x 6*
`The answer is:` `360`

-------------------------------------------------------------------------------------------------

**Illustration of diffusion intermediate process when the generative trajectory starts to be separated**

token index    1     2    3   4    5 6 7   8   9   10   11
**Baseline:** `Lorie earns $10 per hour,` ⊗ ⊗ ⊗ ⊗ ⊗ **...** *Step #7*

↓*diffusion, do sampling procedure in Eq.(1)*

`Top-1 confident index of single-step predictive distributions:` `7th mask token`

↓*decoding*

`Lorie earns $10 per hour, so` ⊗ ⊗ ⊗ ⊗ **...** *Step #8*

---

**Ours:** `Lorie earns $10 per hour, so` `Karen` `earns twice that, which is $10 x 2 =` `$20 per hour.` *The generative trajectory starts to be different from the results of baseline at the "Karen"*

`If Karen works 3 hours per day, she earns $20 x 3 = $60 per day.`
`In two days, Karen earns $60 x 2 = $120.`
`The answer is:` `120`

-------------------------------------------------------------------------------------------------

**Illustration of diffusion intermediate process when the generative trajectory starts to be separated**

token index    1     2    3   4    5 6 7   8   9   10   11
**Ours:** `Lorie earns $10 per hour,` ⊗ ⊗ ⊗ ⊗ ⊗ **...** *Step #7*

↓*diffusion, do sampling procedure in Eq.(12)*

`Top-1 confident index of approximated target distributions:` `8th mask token`

↓*decoding*

`Lorie earns $10 per hour,` ⊗ `Karen` ⊗ ⊗ ⊗ **...** *Step #8, reject single-step top-1 confident index*

*Table 8.* Final outputs of different sampling procedures on an example of the GSK8K benchmark. We present the reason analysis of the different final outputs by tracking the trajectory separated points.

**Task:** *Kylar went to the store to buy glasses for his new apartment. One glass costs $5, but every second glass costs only 60% of the price. Kylar wants to buy 16 glasses. How much does he need to pay for them??* `Incorrect`
`Correct` ⊗ Mask token

**Baseline:** `Kylar wants to buy 16 glasses, so he needs to pay for 16 glasses.`
`The first glass costs $5.`---→ *The generative trajectory starts to be different from our results at the "."*
`The second glass costs 60% of the price, which is $5 *0.6 = $3.`
`The third glass costs $5.`
`The fourth glass costs 60% of the price,whichis $5*0.6 =$3The fifth glass`
`costs $5.The sixth glass costs 60%of the price,which is $5 *0.6 = $3.`
`The seventh glass costs $5.`
`The eighth glass costs60% of the price,which is $5* 0.6 = $3.`
`The ninth glass costs $5.`
`The tenth glass costs60% of the price,which is $5 *0.6 = $3.`
`The eleventh glass costs $5.`
`The twelfth glass costs 60% of the price, which is $5 * 0.6 = $3.The`
`thirteenth glass costs $5.The fourteenth glass costs 60% of the price,`
`which is $5 * 0.6 = $3.The fifteenth glass costs $5.The sixteenth`
`glass costs 60% of the price, which is $5 * 0.6 = $3.Kylar needs to`
`pay$5*8+$3*8=$40+$24=$64.`
`The answer is:` `64`

- - - - - - - - - - - - - - - - - - - - - - - - - - - - - - - - - - - - - - - - - - - - - - - - -

**Illustration of diffusion intermediate process when the generative trajectory starts to be separated**

token index   20  21  22  23  24 25 26 27 28 29
**Baseline:** `...` `The first glass costs $5` ⊗ ⊗ ⊗ ⊗ ⊗ `...` - - -→ *Step #25*

↓*diffusion, do sampling procedure in Eq.(1)*

`Top-1 confident index of single-step predictive distributions:` `25th mask`
`token`

↓*decoding*

`...` `The first glass costs $5.` ⊗ ⊗ ⊗ ⊗ `...` - - - -→ *Step #25*

**Ours:** `Kylar wants to buy 16 glasses, so he needs to pay for 16 glasses in`
`total.`
`The first glass costs $5, and every second glass costs 60% of the price,`
`which is $5 * 60/100 = $3. So, every second glass costs $3.`
`Since Kylar wants to buy 16 glasses, he will have to pay for 8 glasses at`
`the regular price of $5 and 8 glasses at the discounted price of $3.`
`The cost of 8 glasses at the regular price is 8 * $5 = $40.`
`The cost of 8 glasses at the discounted price is 8 * $3 = $24.`
`Therefore, Kylar needs to pay $40 + $24 = $64 for the 16 glasses.`
`The answer is:` `64`

- - - - - - - - - - - - - - - - - - - - - - - - - - - - - - - - - - - - - - - - - - - - - - - - -

**Illustration of diffusion intermediate process when the generative trajectory starts to be separated**

token index   20  21  22  23  24 25 26 27 28 29
**Baseline:** `...` `The first glass costs $5` ⊗ ⊗ ⊗ ⊗ ⊗ `...` - - -→ *Step #25*

↓*diffusion, do sampling procedure in Eq.(12)*

`Top-1 confident index of approximated target distributions:` `28th mask`
`token`

↓*decoding*

`...` `The first glass costs $5` ⊗ ⊗ ⊗ `second` ⊗ `...` -→ *Reject single-step top-1 confident index*

*Table 9.* Final outputs of different sampling procedures on an example of the GSK8K benchmark. We present the reason analysis of the different final outputs by tracking the trajectory of the separated points.

> **Task:** *You plan to visit 4 European cities for 22 days in total. You only take direct flights to commute between cities. You would like to visit Warsaw for 6 days. You would like to meet your friends at Warsaw between day 1 and day 6 to tour together. You would like to visit Vienna for 6 days. You want to spend 6 days in Mykonos. You want to spend 7 days in Vilnius. Here are the cities that have direct flights: Warsaw and Vilnius, Warsaw and Vienna, Vilnius and Vienna, Vienna and Mykonos. Find a trip plan of visiting the cities for 22 days by taking direct flights to commute between them. SOLUTION:* `Incorrect` `Correct` ⊗ Mask token

**Baseline:**           Here is the trip plan for visiting the 4 European cities for 22
days:
\*\*Day 1-6:\*\* Arriving in Warsaw and visit Warsaw for 6 days.
\*\*Day 6:\*\* Fly from Warsaw to Vilnius.
\*\*Day 6-12:\*\* Visit Vilnius for 7 days.
\*\*Day 12:\*\* Fly from Vilnius to Vienna.
\*\*Day 12-1`8`:\*\* Visit Vienna for 6 days. ----⟩ *It should be 7 instead of 8 to match the 6 days*
\*\*Day 1`8`:\*\* Fly from Vienna to Mykonos.---------⟩ *Accumulated error from the last line*
\*\*Day 1`8`-22:\*\* Visit Mykonos for 6 days.---------⟩ *Accumulated error from the last line*

--------------------------------------------------------------------------------
**Illustration of diffusion intermediate process when three mask tokens left**

**Baseline:**           Here is the trip plan for visiting the 4 European cities for 22
days:
\*\*Day 1-6:\*\* Arriving in Warsaw and visit Warsaw for 6 days.
\*\*Day 6:\*\* Fly from Warsaw to Vilnius.
\*\*Day 6-12:\*\* Visit Vilnius for 7 days.
\*\*Day 12:\*\* Fly from Vilnius to Vienna.
\*\*Day 12-1`8`:\*\* Visit Vienna for 6 days. -⟩ *Semantically ambiguous token was decoded early.*
\*\*Day 1`8`:\*\* Fly from Vienna to Mykonos.
\*\*Day 1`8`-22:\*\* Visit Mykonos for ⊗ ⊗ ⊗----⟩ *Left last three mask tokens at the end of sentence*

**Ours:** Here is the trip plan for visiting the 4 European cities for 22 days:
\*\*Day 1-6:\*\* Arriving in Warsaw and visit Warsaw for 6 days.
\*\*Day 6:\*\* Fly from Warsaw to Vilnius.
\*\*Day 6-12:\*\* Visit Vilnius for 7 days.
\*\*Day 12:\*\* Fly from Vilnius to Vienna.
\*\*Day 12-1`7`:\*\* Visit Vienna for 6 days.---------⟩ *Correct*
\*\*Day 1`7`:\*\* Fly from Vienna to Mykonos.---------⟩ *Correct*
\*\*Day 1`7`-22:\*\* Visit Mykonos for 6 days.---------⟩ *Correct*

--------------------------------------------------------------------------------
**Illustration of diffusion intermediate process when three mask tokens left**

**Ours:** Here is the trip plan for visiting the 4 European cities for 22 days:
\*\*Day 1-6:\*\* Arriving in Warsaw and visit Warsaw for 6 days.
\*\*Day 6:\*\* Fly from Warsaw to Vilnius.
\*\*Day 6-12:\*\* Visit Vilnius for 7 days.
\*\*Day 12:\*\* Fly from Vilnius to Vienna.
\*\*Day 12-1 ⊗ :\*\* Visit Vienna for 6 days.-⟩ *Semantically ambiguous token maintains masked.*
\*\*Day 1 ⊗ :\*\* Fly from Vienna to Mykonos.-⟩ *Semantically ambiguous token maintains masked.*
\*\*Day 1 ⊗ -22:\*\* Visit Mykonos for 6 days.-⟩ *Semantically ambiguous token maintains masked.*

*Table 10.* Upper: final outputs of different sampling procedures on an example of the Trip benchmark. Bottom: The reason analysis of the different final outputs by tracking the late-stage generative results (three mask tokens are left).

