# OpenReview forum: "Beyond Confidence: Adaptive and Coherent Decoding for Diffusion Language Models"
_ICML.cc/2026/Conference — ICML 2026 regular_

### Official Review · Reviewer_qxgp · 2026-03-10

**Soundness:** 3
**Presentation:** 2
**Significance:** 3
**Originality:** 3
**Overall Recommendation:** 4
**Confidence:** 4

**Summary:**

Diffusion Language Models (DLMs) unmask tokens iteratively, and existing decoders select which positions to unmask based on single-step confidence alone.

This paper argues that single-step confidence is an unreliable proxy for the true marginal token distribution, because the prediction at any position depends on which surrounding tokens have already been decoded, and that context is still changing. The proposed fix, CCD, maintains a short rolling buffer of recent predictive distributions and selects positions whose predictions are consistent across those steps rather than merely confident at the current one. An adaptive variant, CCD-DS, uses this consistency signal to expand the per-step decoding budget dynamically. The method requires no retraining, works with existing DLM checkpoints, and is evaluated on Dream-7B-Instruct and LLaDA-8B-Instruct across math, code, and planning benchmarks. CCD shows modest but consistent gains under standard single-token-per-step decoding and substantially better robustness when multiple tokens are decoded per step.

**Compliance With Llm Reviewing Policy:**

Affirmed.

**Key Questions For Authors:**

1> Please provide complete pseudocode for CCD and CCD-DS in the main paper. What happens when the coherent candidate set is empty? When it contains fewer positions than the current decoding budget?

2> Can the authors explicitly characterize what the sliding-window, top-V, intersection approximation loses relative to the idealized trajectory-averaged estimator, and under what conditions the two should be expected to agree?

3> Which parts of the theoretical analysis constitute formal guarantees for the implemented algorithm, and which are intended as motivation or intuition? The current phrasing does not draw this line clearly enough.

4> Can the authors add an ablation separating the individual algorithmic choices: averaging only, averaging with top-V, averaging with top-V and intersection, full CCD, so that it is clear which component is responsible for the reported gains?

**Limitations:**

No.

The paper should acknowledge  the gap between the idealized estimator and the buffer-based implementation, the potential brittleness of intersection-based selection when V is small, possible failure modes when stable predictions are nonetheless wrong, and the absence of direct runtime evidence.

**Strengths And Weaknesses:**

Strength:

1> The high-parallelism results in Table 2 are the paper's strongest contribution. On HumanEval with bt=3, the standard decoder falls to 23.17% while CCD holds at 45.73%. That is impressive result; it directly validates the paper's core claim that single-step confidence is structurally fragile under increased decoding budget.

2> The method is training-free, compatible with both entropy-based and max-probability selection, and adds only a small rolling buffer. The practical barrier to adoption is low.

3> The mutual information framing in Proposition 3.1 is a clean decomposition. The idea that tokens with high I(x_i; c | s) are semantically hard and should be deferred is intuitive and consistent with the case studies in Tables 6–8.

4> CCD-DS produces meaningful efficiency gains without obvious quality loss, particularly on planning tasks. On Dream + Trip, it reaches 19.01% vs 16.93% for fixed-step CCD while using 75 steps instead of 256.

---

> ### Author Rebuttal · Authors · 2026-03-30
>
> ---
> ## KQ 1-1: pseudocode
>
> Please see the attached links:
>
> - CCD: https://files.catbox.moe/ujvxos.jpeg
> - CCD-DS: https://files.catbox.moe/5snwqt.jpeg
>
> **We will provide these pseudocodes into the main paper.**
>
> ---
>
> ## KQ 1-2: Empty coherent set
>
> When the coherent candidate set is empty, CCD falls back to **single-step confidence decoding**: it selects positions using only the current step’s predictive distributions. This corresponds to **Lines 34–35** in Algorithm 1 (CCD) and **Lines 28–30** in Algorithm 2 (CCD-DS), where the fallback picks the top-*b* positions with the lowest entropy *at the current step*.
>
> ---
>
> ## KQ 1-3: Candidates < budget
>
> As in KQ 1-2, CCD uses single-step confidence to **fill any unused budget** when the coherent set is smaller than *b*.  All coherent candidates are decoded from the marginalized distribution, and the rest of the budget goes to the lowest–single-step-entropy positions among the **remaining** masked positions.
>
> CCD-DS behaves differently: it decodes *only* the coherent candidates and does **not** top up the budget, so ambiguous positions can leverage richer later context.
>
> ---
>
> ## KQ 2-1 & L1: Ideal vs. buffer estimator
>
> We agree that the sliding-window top-*V* intersection entails **three approximations** relative to averaging over the full trajectory. We summarize them as follows:
>
> **(1) Window truncation:** Although later decoding steps may produce more informative contexts due to the reveal of more tokens, it is a trade-off to balance the efficiency and effect. First,  the d most recent steps may sufficiently capture the high-quality predictions, and early-step noisy predictions are excluded. Second, it may be computationally inefficient to recompute the marginal entropy of unmasked tokens using more contexts, leading to a more sophisticated decoding procedure.
>
>
> **(2) Top-*V* filtering:** Retaining only the V most confident predictions per step, filtering out low-confidence noise that would reduce the candidate set of marginal estimates of tokens. However, it is also a trade-off to balance the efficiency and effect.
>
> **(3) Intersection:** Requiring consistency in top-*V* across all *d* steps implements the “high confidence *and* high consistency” idea motivated by our MI analysis.
>
> The **buffer-based** estimator should track the **idealized** one well when: (a) d is appropriate to capture the prediction stabilization dynamics, (b) V captures the true confident set, and (c) the base model's predictions are sufficiently calibrated.
>
> **We will add this characterization to the revision.**
>
>
> ---
>
> ## KQ 3: Guarantees vs. intuition
>
> One of the most important parts is **Proposition 3.1 (MI decomposition)** , which shows that marginal entropy equals conditional entropy plus conditional mutual information. It therefore provides **theoretical motivation** for CCD, which approximates marginal-entropy–based decisions in a practical way.
>
> The analysis around **Eq. (9)** further links the decomposed conditional mutual information to **bounding sampling error**.
>
> **We will integrate this discussion in the revised paper.**
>
> ---
>
> ## KQ 4: Ablations
>
> We compare the following variants of CCD:
>
> - **Averaging only:** Marginal entropy averaged over *d* steps for **all** masked positions (no per-step top-*V* or intersection).
> - **Averaging with top-*V*:** Marginal entropy only for positions in each step’s top-*V*; the number of distinct positions considered ranges from *V* to *dV*, and the number of predictive distributions per position ranges from *d* down to 1.
> - **Averaging with top-*V* and intersection:** This matches **CCD**.
>
> | Method | GSM8K Accuracy |
> |:---|:---:|
> | Default (Dream) | 81.01 |
> | + Averaging only | 81.90 |
> | + Averaging with top-V | 80.39 |
> | + Averaging with top-V and intersection (**CCD**) | **82.51** |
>
> As the table shows, averaging-only and CCD improve over default single-step decoding, but **averaging with top-*V* alone** hurts accuracy. This is plausible because:
>
> 1. **Unstable marginal estimates:** A token that enters top-*V* only at an early step can still contribute to the average even when it is weakly aligned with the **current** context, which hurts decoding.
>
> 2. **Noise vs. selection:** The averaging-only baseline uses every masked position every step, which admits noisy predictions. CCD’s **intersection across steps** is a stricter filter and yields a more reliable, cheaper marginal-entropy proxy.
>
> ---
>
> ## Limitations
>
> We **will include** a limitation section in the revised manuscript as follows:
>
> - **Idealized vs. buffer-based estimators**: Please see the discussion in the response to KQ 2-1 & L1.
>
> - **The potential brittlen ... when V is small:** Keeping only the *V* most confident positions per step suppresses low-confidence noise in the marginal estimate. In the future, it may be interesting to explore a dynamic buffer size for better robustness.
>
> - **Runtime:** We will discuss this point by the reported FPS as in the Table @ Reviewer iEHV.

---

> > ### Author Rebuttal · Reviewer_qxgp · 2026-04-01
> >
> > I deemed author’s responses adequate and appreciate author’s commitment on enhancing the new revision.

---

> > > ### Author Response · Authors · 2026-04-07
> > >
> > > We sincerely appreciate your feedback! Your comments have enhanced the quality of our paper.
> > >
> > >
> > > Thank you again!
> > >
> > > The authors

---

### Official Review · Reviewer_iEHV · 2026-03-12

**Soundness:** 4
**Presentation:** 4
**Significance:** 3
**Originality:** 4
**Overall Recommendation:** 4
**Confidence:** 3

**Summary:**

This paper identifies limitations in the single-step sampling approach used during DLMs inference and proposes a decoding method that comprehensively leverages historical contextual states to evaluate prediction confidence. Through detailed theorem and formula derivations, the paper demonstrates the suboptimality of single-step sampling and the validity of utilizing predictions from previous steps . The corresponding implementable decoding strategy CCD is designed, and its effectiveness is validated through experiments.

**Compliance With Llm Reviewing Policy:**

Affirmed.

**Key Questions For Authors:**

As highlighted in the Weaknesses section, I recommend the authors address the following points:

1. Provide the hyperparameter settings for Table 3 and analyze how this setup enables CCD-DS to match fast-dLLM's speed while maintaining better performance.
2. Analyze the computational overhead introduced by the CCD operations and report its actual impact on inference latency.
3. Provide an ablation study on the historical iteration step parameter d and justify the rationale behind the default choice.
4. Provide a clearer comparative analysis with dInfer to explicitly highlight the unique contributions of this work.

**Limitations:**

The author should include a discussion of the paper's limitations.

**Strengths And Weaknesses:**

**Strengths**

The paper exhibits a solid theoretical motivation. Rather than just proposing a heuristic, the authors mathematically derive the limitations of current single-step sampling, which naturally sets up their proposed solution. The implementation is straightforward, highly plug-and-play, and visibly improves generation quality. Finally, the paper proposes an adaptive acceleration version CCD-DS, which further enhances its value.

**Weaknesses**

1. Since dInfer (evaluated as "credit decoding" in the experiment) previously explored using historical states to improve sampling, the paper would benefit from a clearer discussion on CCD's unique contributions.
2. The current evaluation mainly relies on the reduction of "inference steps" to demonstrate speedups. However, the CCD framework introduces its own computational operations. It is unclear how this additional overhead affects the actual inference latency.
3. The authors state that the parameter settings for Table 3 are provided in the appendix, but they appear to be missing.It is unclear how CCD-DS matches the speed of fast-dLLM while achieving better performance.
4. The paper currently lacks an ablation study on the historical iteration step parameter d.

---

> ### Author Rebuttal · Authors · 2026-03-30
>
> ---
> ## W1 & KQ 4: Differentiation from dInfer (credit decoding)
>
> - One key difference is that CCD derives its candidate selection criterion from **conditional mutual information** (Prop. 3.1), which connects to a **sampling error bound** (Prop. 3.3). In contrast, dInfer does not provide an equivalent theoretical formulation for why its credit-fusion mechanism leads to better decoding decisions.
>
> - Second, dInfer continuously fuses logits using **long-term credit scores**, which may have limited effectiveness in long-context settings and can introduce over-smoothing. CCD instead uses the **intersection of top-*V* confident sets** across the last *d* steps, explicitly targeting **predictive consistency**: tokens are prioritized only when they are consistently confident under multiple recent contexts.
>
> **We will provide the detailed analysis in the revised version.**
>
> ---
> ## W2 & KQ 2: Computational overhead and inference latency
>
> $\color{red}{\textbf{Computational overhead analysis:}}$ CCD adds computation after the standard forward pass. Specifically, at each decoding step it performs post-processing on the output logits: (i) computing entropy for candidates in the per-step top-*V* set, (ii) intersecting top-*V* sets across the last *d* steps, and (iii) constructing the marginalized distributions for the selected positions. Since these are **post-logit operations**, the additional overhead is typically small relative to the model forward cost and can be combined with common acceleration strategies (e.g., KV-caching).
>
> $\color{red}{\textbf{Inference latency analysis:}}$ We report the **peak GPU memory**, **FPS** (as an empirical proxy for inference latency), and **accuracy on GSM8K** using the dInfer evaluation engine. The results show that CCD achieves comparable throughput to the baseline while improving accuracy:
>
> **Table 1**
> | Method | GSM8K Acc | Avg. GPU Memory | FPS |
> |:---|---:|---:|---:|
> | Default (threshold confidence) | 89.3 | 9.43 GB | 239.2 |
> | dInfer (credit decoding) | 89.2 | 9.61 GB | 218.5 |
> | CCD (ours) | 89.8 | 9.58 GB | 220.9 |
>
> These measurements suggest that the CCD-specific operations introduce **only a small FPS change**, and the overall latency is not dominated by the additional buffer maintenance. We further plan to explore additional implementation optimizations to reduce the remaining overhead (e.g., restricting computation to top-*p* logits and using cheaper consistency proxies than full entropy).
>
> **We will provide the detailed analysis in the revised version.**
>
> ---
> ## W3 & KQ 1: Hyperparameter settings for Table 3 (and speed–performance rationale)
>
> For **Fast-dLLM**, the confidence threshold is set to **0.9**, where confidence is computed as the **maximum token probability** from the model distribution.
>
> To match the speed of Fast-dLLM, CCD-DS uses a per-task buffer size *V* (in Table 3) so that the decoder can allocate a larger decoding budget per step when appropriate:
>
> | Dataset | GSM8K | Math | HumanEVAL | MBPP |
> |:---|---:|---:|---:|---:|
> | Buffer size *V* | 8 | 8 | 10 | 10 |
>
> $\color{red}{\textbf{Analysis:}}$  Intuitively, a larger *V* increases the number of token candidates considered for decoding. This can improve accuracy by allowing CCD-DS to include tokens that are not highly confident in a single step but are **predictively consistent** across the last *d* steps. Importantly, CCD-DS does not require extra model forward passes beyond the standard decoding process, so the additional latency remains aligned with the fast-dLLM setting in **Table 3**.
>
> **We will provide the detailed analysis in the revised version.**
>
> ---
> ## W4 & KQ 3: Ablation on historical iteration step *d* and default choice
>
> We conduct an ablation over the historical iteration step parameter *d*, while fixing the buffer size as *V = 4*.
>
> | *d* | GSM8K | Math | HumanEVAL | MBPP | Avg |
> |:---|---:|---:|---:|---:|---:|
> | 0 (degrades to single-step decoding) | 81.01 | 40.90 | 52.66 | 58.0 | 58.00 |
> | 1 | 81.65 | 41.20 | 55.32 | 58.2 | 58.20 |
> | 2 | 82.26 | 40.46 | 57.92 | 58.4 | 58.40 |
> | 3 | 82.51 | 41.24 | 56.71 | 59.0 | 59.00 |
> | 4 | 81.90 | 40.54 | 57.99 | 56.80 | 56.80 |
>
> $\color{red}{\textbf{Analysis:}}$ The results indicate that *d = 3* provides the best average performance, achieving a strong trade-off: it benefits from longer historical consistency while avoiding the diminishing returns (and potential noise accumulation) observed when *d* grows further.
>
> **We will provide the detailed analysis in the revised version.**
>
> ---
>
> ## Limitations
>
> We **will include** a limitation section in the revised manuscript as follows:
>
> - **Idealized vs. buffer-based estimators**: Please see the discussion in KQ 2-1 @ Reviewer qxgp.
>
> - **The potential brittleness when V is small:** In the future, it may be interesting to explore a dynamic buffer size for better robustness.
>
> - **Runtime:** We will discuss this point by the reported FPS, as in the Table 1.

---

> > ### Author Rebuttal · Reviewer_iEHV · 2026-04-02
> >
> > Thank you for the detailed response, my concerns have been addressed, and I will maintain my current score.

---

> > > ### Author Response · Authors · 2026-04-07
> > >
> > > Thanks for your feedback!
> > >
> > > The authors

---

### Official Review · Reviewer_8s5n · 2026-03-12

**Soundness:** 2
**Presentation:** 2
**Significance:** 3
**Originality:** 3
**Overall Recommendation:** 3
**Confidence:** 4

**Summary:**

This paper proposes Coherent Contextual Decoding (CCD), a training-free decoding framework for Diffusion Language Models (DLMs) that attempts to improve upon single-step confidence-based sampling by leveraging historical contexts to approximate marginal distributions and modeling consistency via conditional mutual information. The authors also propose an adaptive variant, CCD-DS, to dynamically adjust decoding budgets. Experiments are conducted on LLaDA-8B and Dream-7B (early 2025 baselines) across mathematical reasoning, code generation, and planning tasks, showing modest performance gains.

**Compliance With Llm Reviewing Policy:**

Affirmed.

**Final Justification:**

I have carefully reviewed the authors' responses and appreciate the additional experiments, particularly the dInfer integration and the FPS evaluations for the LLaDA models.

However, the complete accuracy-latency trade-off remains unclear. The manuscript has shown that CCD achieves a more significant accuracy gain on the Dream-7B model. Unfortunately, explicit latency evaluations for Dream-7B are currently missing due to framework integration issues. Without seeing the exact efficiency cost for the model that delivers the best performance, it is difficult to fully assess the practical value of this method.

Therefore, I maintain my recommendation of weak reject. I highly encourage the authors to integrate the Dream architecture into the dInfer framework to demonstrate these trade-offs in future submissions explicitly.

**Key Questions For Authors:**

1. Compatibility with KV-cache and high-parallelism systems: CCD requires maintaining the Historical Buffer $\mathcal{H}_t$ to store historical probability distributions, creating potential memory conflicts with KV-cache-based acceleration techniques like Fast-dLLM. Have you tested CCD's actual memory footprint and inference latency when KV-cache is enabled? For native high-parallelism block diffusion architectures (e.g., d2f-LLaDA/Dream, SDAR), does CCD's historical dependency mechanism (requiring waiting $d$ steps to accumulate context) cancel out the parallelism advantages of these architectures? Table 2 shows significant performance degradation even at $b_t=3$; does this indicate that CCD is fundamentally ill-suited for highly parallel decoding scenarios?

2. Regarding model timeliness: Your method is validated on LLaDA-8B and Dream-7B (early 2025 models). Given the rapid recent development of diffusion language models (e.g., larger-scale commercial models or improved training methods), do you have plans to validate CCD on newer model architectures or larger parameter scales?

**Strengths And Weaknesses:**

**Strengths:**

The idea of utilizing historical prediction consistency to guide decoding is somewhat novel, and the conditional mutual information perspective provides a new analytical dimension for understanding diffusion sampling.


**Weaknesses:**

1. Incompatibility with state-of-the-art (SOTA) inference optimizations: CCD requires maintaining a historical probability buffer of size $d \times V \times |\mathbb{X}|$, which conflicts with memory management strategies in KV-cache-based acceleration systems like Fast-dLLM.

2. Although DLM decoding optimization is an important problem, the practical value of this work is highly limited and diverges from the field's development direction. The current core demands in DLM inference are latency reduction and memory efficiency. The mainstream industry and academic directions focus on KV-cache optimization (e.g., Fast-dLLM) and native high-parallelism block diffusion architectures (e.g., d2f-LLaDA/Dream, SDAR).

CCD introduces additional memory overhead (storing historical probability distributions) and cross-step state dependencies (requiring accumulation of $d$ steps of history) to achieve "historical consistency," which creates fundamental tensions with these directions:
- Conflict with KV-cache systems: CCD requires maintaining $\mathcal{H}_t$ to store historical distributions, which is difficult to reconcile efficiently with memory management strategies in Fast-dLLM-like systems;
- Conflict with modern inference engines: Although claimed to be plug-and-play，engines like SGLang and vLLM are deeply optimized for stateless/minimal-state decoding loops (e.g., PagedAttention, Continuous Batching). CCD requires intrusive modifications to the scheduling logic to support cross-step state accumulation, buffer updates, and distribution aggregation (Eq. 12), introducing substantial engineering integration complexity;
- Conflict with native high-parallelism architectures: Table 2 shows significant performance degradation for CCD even at $b_t=3$ (e.g., MATH drops from 41.24% to 35.68%), indicating difficulty adapting to true block diffusion high-parallelism scenarios.

3. While the paper structure is logically organized, the execution contains obvious deficiencies. Figure 2 suffers from crowded layout with text exceeding boundary boxes; Table 4 exhibits low-resolution images that impede understanding of critical diffusion trajectories. This neglect of visual presentation details reflects a lack of rigor in paper preparation and undermines the credibility of technical contributions.

---

> ### Author Rebuttal · Authors · 2026-03-30
>
> We highly appreciate and acknowledge your detailed suggestions, especially regarding the compatibility of CCD with current inference and acceleration strategies.
>
> ---
> ## W1 & KQ 1: KV-cache compatibility and memory footprint
> -  We respectfully clarify that $\color{blue}{\textbf{CCD is NOT inherently incompatible with KV-cache–based acceleration}}$. Specifically, **KV-cache** speeds up the **forward pass**; CCD only changes **post-forward token selection** using logits. $\color{blue}{\textbf{The two can coexist without conflict}}$.
>
> -  We respectfully clarify that the historical buffer stores $\color{blue}{\textbf{probability distributions for a small set of candidate positions}}$ (top-*V* over the last *d* steps), $\color{blue}{\textbf{not attention states}}$. Specifically, CCD’s buffer is comparatively lightweight: roughly **O(*B*·*d*·*V*)** per step for the stored distributions, rather than the overall sequence predictions (with *d*·*V* ≪ sequence length *L* in typical settings).
>
> - As an implementation, $\color{blue}{\textbf{we have integrated the CCD into the recent high-parallelism inference engine, i.e., dInfer}}$, which enables variable acceleration strategies (e.g., $\color{blue}{\textbf{vLLM, SGLang, and KV-cache optimization}}$). Please see the code in **https://anonymous.4open.science/r/dInfer-CCD-3D23/**  and inference results in Table 1, where we report **peak GPU memory**, **FPS** (forward pass per second, as a proxy for inference latency), and **GSM8K accuracy** using the **dInfer** engine with standard acceleration options (including KV-cache–style optimizations where applicable):
>
> **Table 1**
> | Method (LLaDA 2.0 Mini) | GSM8K Acc |  GPU Memory | FPS |
> |:---|---:|---:|---:|
> | Default (threshold confidence) | 89.3 | 9.43 GB | 239.2 |
> | Credit (dInfer) | 89.2 | 9.61 GB | 218.5 |
> | CCD (ours) | 89.8 | 9.58 GB | 220.9 |
>
> These results indicate that CCD maintains a practical memory–latency level  relative to strong baselines under the same engine.
>
> **We will provide the detailed analysis and code in the revised version.**
>
> ---
>
> ## W2 & KQ 2: Modern inference engines (e.g., vLLM) and engineering integration
>
>  - We respectfully clarify that $\color{red}{\textbf{CCD can be compatible with modern inference engines.}}$
>
> - As an example, we have integrated the CCD into the recent high-parallelism inference engine, i.e., **dInfer**, which enables variable acceleration strategies. Please see the code in **https://anonymous.4open.science/r/dInfer-CCD-3D23/python/dinfer/decoding/parallel_strategy.py**  and inference results in Table 1.
>
> ---
>
> ## KQ 3: High-parallelism block diffusion (Table 2) and whether CCD is “fundamentally ill-suited.”
>
> - We acknowledge the accuracy drop at **high per-step parallelism** (e.g., **MATH: 41.24% → 35.68%** when the per-step budget is large).  $\color{blue}{\textbf{However, CCD remains the strongest among the compared methods under the same aggressive parallel setting}}$, and its $\color{blue}{\textbf{relative degradation is smaller}}$ than several baselines (e.g., margin-confidence decoding shows a larger drop in the same table). Thus, the observation reflects a **general tension between aggressive parallel unmasking and stable uncertainty estimation**, $\color{red}{\textbf{not a failure mode unique to CCD}}$.
>
> - Moreover, **Table 3** (dynamic budgets / more realistic parallel decoding configurations) shows that CCD can deliver large **efficiency gains** without catastrophic accuracy loss in most cases, $\color{blue}{\textbf{which better reflects practical high-throughput usage}}$. We will articulate this distinction (fixed aggressive *b* in Table 2 vs. dynamic budgets in Table 3) more clearly in the paper.
>
> ---
>
> ## KQ 4: Does CCD “wait” *d* steps and cancel parallelism?
>
> $\color{blue}{\textbf{No mandatory waiting is required.}}$  When fewer than *d* historical steps are available (e.g., early in decoding), CCD **degrades to single-step selection** without blocking the schedule to “collect” *d* full steps. The buffer is **rolling** and **warm-started** naturally as decoding proceeds. Therefore, CCD does not introduce an additional *d*-step **pipeline stall** solely to accumulate history; it adds **state** that can be updated each step incrementally.
>
> ---
>
> ## KQ 5: Presentation (Figure 2 and Table 4)
>
> We will improve Figure 2's layout and increase Table 4's font size ($\color{blue}{\textbf{Note that it is a table, rather than a figure}}$) for better readability in the revised revision.
>
> ---
>
> ## KQ 6: Model timeliness and evaluation on newer / larger models
>
> - We agree that the diffusion-LM landscape evolves quickly. Beyond the most representative **LLaDA-8B** and **Dream-7B** setups in the main paper, we are **actively evaluating CCD on additional architectures and larger scales**.
>
> - As an initial datapoint, we report $\color{red}{\textbf{LLaDA 2.0 Mini}}$ with the dInfer engine in **KQ 1 (above)**; we will make the evaluation roadmap explicit in the revision.

---

> > ### Author Rebuttal · Reviewer_8s5n · 2026-04-01
> >
> > Thank you for providing such a detailed response. I observe that the FPS for CCD is slightly higher than for Credit (dInfer) (220.9 vs. 218.5). This suggests a potential trade-off where inference speed is sacrificed for a minor improvement in accuracy, which I am still concerned about. Additionally, given that LLaDA 8B and Dream 7B are the primary models featured in your paper, why are FPS experiments not evaluated on them?

---

> > > ### Author Response · Authors · 2026-04-02
> > >
> > > We thank you for your follow-up comments. Below we provide point-by-point responses.
> > >
> > > ---
> > >
> > > ## Q1: I observe that the FPS for CCD is slightly higher than for Credit (dInfer) (220.9 vs. 218.5). This suggests a potential trade-off where inference speed is sacrificed for a minor improvement in accuracy, which I am still concerned about.
> > >
> > >
> > > - *"I observe that the FPS for CCD is slightly higher than for Credit (dInfer) (220.9 vs. 218.5)."*: $\color{blue}{\textbf{We would like to clarify that CCD’s higher FPS (220.9 vs. 218.5 with Credit) actually corresponds to lower latency, as FPS measures the number of forward passes per second.}}$ Therefore, **CCD has a joint improvement in terms of effectiveness and efficiency comapred with Credit.** This is reasonable, as CCD’s buffer is comparatively lightweight and avoids computing over the full logit space.
> > >
> > > - *"a minor improvement in accuracy"*: $\color{blue}{\textbf{For the degree of the accuacy improvement, we would like to clarify that it is more reasonable to consider more tasks and decoding scenarios:}}$
> > >
> > > 1. As shown in the following table, our proposed method enjoys a moderate accuracy gain with improved efficiency (CCD-DS here) on more tasks.
> > >
> > > @ Table 1 (Dream-7B-instruct) in the original paper
> > > | Method | HumanEVAL Acc | Trip Acc | Math Acc |
> > > |:---|---:|---:|---:|
> > > | Default (entropy) | 52.66 | 15.10 | 40.90 |
> > > | Ours | **56.61** | **19.01** | **41.20** |
> > >
> > > 2. As shown in the following table, our proposed method enjoys an **obvious** accuracy gain on fixed high decoding budgets.
> > >
> > > @ Table 2  (Dream-7B-instruct) in the original paper,
> > > | Method ($b_{t}=3$) | HumanEVAL Acc | GSM8K Acc | Math Acc |
> > > |:---|---:|---:|---:|
> > > | Default (entropy) | 23.17 | 69.07 | 22.87 |
> > > | Ours | **45.73** | **78.55** | **30.12** |
> > >
> > > - We would also acknowledge that, like most existing advanced decoding algorithms, CCD indeed incurs additional computation in logit space compared to naive decoding. **However, CCD has an efficiency advantage **(CCD: 220.9 vs. Credit: 218.5 for FPS)** over counterparts due to its simple and lightweight buffer design and computation.**
> > >
> > > - While inference latency is important, we would highlight that **one of the most challenging issues for modern DLMs may be severe accuracy degradation in high-parallel decoding settings.**  Notably, as shown in Tables 2 and 3 (see https://files.catbox.moe/m6v9q2.jpeg) of the submitted paper, we observe that CCD can deliver large efficiency gains without catastrophic accuracy loss in most cases, which better reflects the balance between efficiency and effectiveness.
> > >
> > > ---
> > >
> > > ##  Q2: Why are FPS experiments not evaluated on LLaDA 8B and Dream 7B?
> > >
> > >
> > > - $\color{blue}{\textbf{The modern inference engine, i.e., dInfer, does not support Dream-7B for acceleration and does not support SGLang for LLaDA-8B.}}$ Following your last-round comments, you were mainly concerned about *whether CCD can be compatible with existing inference engines when using acceleration strategies (e.g., SGLang).* To address your concern, we use dInfer as our inference backend, which enables SGLang and KV-cache acceleration with fair latency measurement. **However, dInfer does not support Dream-7B, and LLaDA-8B cannot be run with SGLang on dInfer.** $\color{blue}{\textbf{Therefore, we mainly report FPS on the LLaDA-2.0 series (which can be compatible with SGLang and KV-cache acceleration), rather than on Dream-7B and LLaDA-8B.}}$
> > >
> > > - $\color{blue}{\textbf{Additional results on LLaDA-8B}}$: We provide FPS results for different decoding algorithms on LLaDA-8B, which only supports naive acceleration strategies (e.g., tensor parallelism and prefix/dual KV-cache acceleration) on dInfer without SGLang (hence lower FPS than with SGLang). As illustrated in the following tables, CCD can achieve better performance compared with Credit and default threshold decoding, e.g., **from 33.9 (Credit) to 36.2 (CCD)**. Meanwhile, like most existing decoding algorithms, CCD and Credit incur additional computation in logit space compared to naive decoding. However, because CCD’s buffer is comparatively lightweight, CCD achieves lower inference latency (higher FPS) than counterparts, e.g., **82.4 (CCD) vs. 80.2 (Credit) on GSM8K.**
> > >
> > >
> > >
> > > *LLaDA-8B on HumanEVAL using dInfer with a threshold of 0.9*
> > > | Method | HumanEVAL Acc | GPU Memory | FPS ↑ |
> > > |:---|---:|---:|---:|
> > > | Default (threshold confidence) | 34.3 | 8.56 GB | 67.8 |
> > > | Credit (dInfer) | 33.9 | 8.62 GB | 62.4 |
> > > | CCD (ours) | **36.2** | 8.59 GB | 65.2 |
> > >
> > >
> > > *LLaDA-8B on GSM8K using dInfer with a threshold of 0.9*
> > > | Method | GSM8K Acc | GPU Memory | FPS ↑ |
> > > |:---|---:|---:|---:|
> > > | Default (threshold confidence) | 78.22 | 6.95 GB | 86.1 |
> > > | Credit (dInfer) | 78.12 | 6.99 GB | 80.2 |
> > > | CCD (ours) | **78.85** | 6.97 GB | 82.4 |
> > >
> > >
> > >
> > > - For the Dream series, we will try to integrate Dream’s architecture into the dInfer framework and measure inference latency in future work.

---

### Official Review · Reviewer_Btmy · 2026-03-12

**Soundness:** 3
**Presentation:** 1
**Significance:** 3
**Originality:** 3
**Overall Recommendation:** 4
**Confidence:** 3

**Summary:**

The authors propose 2 new Diffusion Language Model (DLM) inference strategies: Coherent Contextual Decoding (CCD) and an adaptive sampling version (CCD-DS). The goals of these 2 methods are to perform faster (i.e. fewer iterations) and more accurate inference than the baseline of a uniform sampling budget (i.e. unmask the same number of tokens every iteration). The key of CCD is that it estimates entropies of the marginal probabilities of the tokens across iterations to identify the token(s) with the lowest entropy. The intuition behind this is that tokens with higher mutual information will be unmasked later than in the baseline inference strategy, which provides more context for the token in order for it to be more accurate. That will also help future tokens be more correct due to the fact that if a token is chosen that has a high mutual information value but is predicted incorrectly, that would set the decoding on the wrong path and lead to much worse decoding results for the remainder of the example. The authors also propose CCD-DS, which dynamically chooses how many tokens to decode each iteration, which leads to faster decoding with minimal degradation in evaluation dataset accuracy. In fact, there were some datasets where no degradation in accuracy was found, and Trip Plan even saw accuracy improvements with CCD-DS over CCD. While the bulk of the results used a baseline of 1-sample uniform sampling, there were also some results comparing the baseline with CCD with 2-token and 3-token decoding per iteration. While all models degraded when increasing the decoding parallelism, CCD decoding degraded less, so the accuracy gap between the baselines and CCD increased as parallelism increased.

The authors also presented further theoretical analysis of CCD, including how the CCD sampling can be thought of as Monte Carlo sampling, a more in-depth discussion on how CCD relates to mutual information, and error bound analysis for decoding time.

Finally, the authors discuss how to practically implement calculating the marginal distributions over iterations via a sliding window and only storing only the top-V most confident token predictions.

**Compliance With Llm Reviewing Policy:**

Affirmed.

**Final Justification:**

I have bumped up my recommendation from 3 to 4 based on the fixes outlined in the rebuttal.

**Key Questions For Authors:**

1. I would have liked to see more discussion on Figure 4a. Why does increasing buffer size lead to fewer inference steps? Is this with CCD or CCD-DS? Do we see similar behavior for the other datasets? Why / why not?
2. In Figure 4a, why is the score so different from the scores we see in Table 1?
3. “On the Trip, the Dream with CCD-DS achieves a score of 19.01% vs. 16.93% with CCD.” I would have liked to see a deeper analysis on this point, perhaps some examples or hypotheses why it’s happening.

I am marking this as a “weak reject” right now because of the poor presentation that I had mentioned. If the revised paper’s improved significantly in its clarity, then I would change my recommendation to an accept. But I cannot in good conscience recommend a paper with so many grammar and other issues related to clarity and spelling.

**Limitations:**

Yes

**Strengths And Weaknesses:**

Strengths:
* CCD and CCD-DS not just presented, but were discussed thoroughly and from different angles. The paper was more than just “here’s an idea we tried and it worked”, but included theoretical justifications regarding why it worked.
* The citations were relevant and sufficient.

Weaknesses
* The primary weakness of the paper is its clarity. There are many awkward sentences and misspelled words.
* There are some parts in the analysis that I would have liked to seen examined further. I have mentioned those in the notes to the authors.

## **Soundness**
* The submission is technically sound. It contains a detailed theoretical analysis as well as some experimental results.
* While some caveats are mentioned, such as efficiency gains rely on the capacity of the base model, the authors focus on their ideas’ strengths and do not discuss weaknesses.

## **Presentation**
* This is the greatest weakness of the paper. The document is hard to follow at times and could benefit from some significant improvements in clarity. There are also spelling errors. Here are just a few examples. There are many more:
    * The authors do not define what the “DS” in CCD-DS stands for.
    * There are some awkward sentences. Examples:
        * “However, existing inference methods typically rely on local, immediate-step metrics, such as confidence or entropy, which inherently lack a more reliable perspective, leading to sub-optimal generation quality.”
            * What do the authors mean by a “reliable perspective”? This is a very difficult sentence to parse, especially when it’s the second sentence of the paper.
        * “Finally, CCD achieves significantly milder performance degradation under highly parallel decoding scenarios compared to baselines.”
            * “Significantly milder” is an unusual and ambiguous term.
        * “Empirically, our method achieves a simultaneous enhancement in both inference speed and performance across diverse benchmarks on Dream and LLaDA.”
            * What do you mean by performance here? Accuracy? Speed? Memory footprint?
        * “We provide a thorough analysis demonstrating that existing sampling procedures utilized the decoding-time single-step predictive distribution as a surrogateof the target distribution is suboptimal.”
            * Awkward sentence.
    * Misspelled words:
        * Table 1: “Planing” -> “Planning”
        * Page 15: “sa ingle” -> “a single”
        * Table 7: “trajectroty” -> “trajectory”

## **Significance**
* Yes, it’s addressing a relevant problem and has broad appeal, as it can be applied to preexisting DLM models as it’s just an inference-time modification.

## **Originality**
* Yes, the work is original.

---

> ### Author Rebuttal · Authors · 2026-03-30
>
> ---
> ## W1, Presentation, clarity, and spelling
>
> We thank the reviewer for the careful reading. Meanwhile, we highly appreciate for pointing out   **specific typos** and  clarity issues, which indeed enhances the presentation of our paper.
>
> Specific modifications and clarifications can be found:
>
> - “Planing” → “Planning”;
>
> - “sa ingle” → “a single”;
>
> - “trajectroty” → “trajectory”
>
> - **Meaning of DS in CCD-DS.** We will make this explicit at first use: **CCD-DS** denotes **CCD with Dynamic Sampling** (adaptive per-step decoding budgets). We will rewrite the early mention on **page 2** (around line 55) for readability, e.g., “**CCD with Dynamic Sampling (CCD-DS)** …”.
>
> - **“A reliable perspective” (introduction).** We intended “reliable” to mean **more stable / trajectory-aware** than purely immediate-step heuristics. We will replace the sentence with a clearer formulation: *“However, existing inference methods typically evaluate generation using local, immediate-step metrics (such as confidence or entropy) rather than a long-term trajectory-aware metric, which inherently leads to sub-optimal generation quality.”*
>
> - **“Significantly milder.”** We agree this phrasing is ambiguous. We will revise this sentence as: *“Finally, CCD exhibits a substantially lower rate of performance degradation under highly parallel decoding scenarios compared to baseline methods.”*
>
> - **“Performance” (Dream / LLaDA).** We will specify **task accuracy** (and, where relevant, **speed / steps** separately) to avoid mixing metrics.
>
> - **Awkward  sentence.** We will rewrite for clarity as: *“We provide a thorough analysis demonstrating that existing sampling procedures are suboptimal, because they use the decoding-time single-step predictive distribution as a surrogate for the true target distribution.”*
>
>
> ## Additional typo/clarity fixes (we found)
>
> - Sec. 4.2 (Main results): “generative **chateristic**” → “generative **characteristics**”.
> - Sec. 4.2 (Main results): “a large **propotion**” → “a large **proportion**”.
> - Sec. 4.3 (Dynamic decoding budgets): “confidence **thereshold**-based” → “confidence **threshold**-based”.
> - Sec. 4.2 (Main results): “more **obviouse**” → “more **obvious**”.
> - Sec. 4 (Datasets): “**ode** Generation” → “**Code** Generation”.
> - Appendix / proofs: “**Details** proofs” → “**Detailed** proofs …”.
>
> $\color{blue}{\textbf{We promise that we will  perform a full pass of proofreading in the revision.}}$
>
> ---
>
> ## W2: Limitation
>
> We will add a limitation section as follows.
>
> - **Idealized vs. buffer-based estimators**: Please see the discussion in KQ 2-1 @ Reviewer qxgp.
>
> - **The potential brittleness when V is small:** In the future, it may be interesting to explore a dynamic buffer size for better robustness.
>
> - **Runtime:** We will discuss this point by the reported FPS.
>
> ---
>
> ## KQ 1
>
> - **(1) Why does increasing buffer size tend to reduce inference steps?** A larger buffer allows CCD-DS to identify **more consistently confident** positions earlier (via the historical top-*V* mechanism), which enables **larger effective per-step decoding budgets** under the dynamic schedule and therefore **fewer total steps** to finish generation. In our Trip ablation, **buffer size 4** achieves a strong trade-off (e.g., high accuracy with substantially fewer steps than the 256-step baseline in the reported setting).
>
> - **(2) CCD or CCD-DS?** Figure 4a is **CCD-DS**, because it concerns **dynamic decoding budgets** (the distinguishing feature of CCD-DS).
>
> - **(3) Do we observe similar behavior on other datasets?** Yes in spirit: **Table 3** shows that modest buffer / schedule adjustments can yield large **efficiency gains** across tasks (Math, coding, etc.), though the exact curves differ with task difficulty and template structure. We will clarify which subplots correspond to which datasets and summarize consistent trends vs. task-specific caveats.
>
> ---
>
> ## KQ 2: Figure 4a vs. Table 1 — why are the scores different?
>
> **Figure 4a** evaluates CCD-DS on a Trip benchmark **subset (City=3 only)**, whereas **Table 1** reports the **full Trip** benchmark (all difficulty levels); this explains the score difference. Although we mentioned this detail, we will highlight it more prominently in the caption.
>
> ---
>
> ## KQ 3: Trip - Dream with CCD-DS  vs. CCD
>
> The Trip Plan benchmark contains substantial template-like content (e.g., "Here is the trip..."), creating long plateau periods of context-insensitive tokens (visible in Figure 3a). We observe that CCD-DS efficiently skips these plateaus by dynamically increasing the decoding budget, which generates template-like content more fast. Therefore, CCD-DS can reduce the risk of potential error accumulation in the generation process of template-like content and focus more steps on semantically ambiguous tokens (e.g., day numbers, city names, see Table 8) in a relatively completed context. These two perspectives contribute to the accuracy improvement of CCD-DS.

---

> > ### Author Rebuttal · Reviewer_Btmy · 2026-04-06
> >
> > Thank you for rebuttal. I have bumped up my overall recommendation.

---

> > > ### Author Response · Authors · 2026-04-06
> > >
> > > We sincerely appreciate your continued support, recognition, and valuable feedback! Your comments have helped improve the quality of our paper, and we are truly grateful for the time and effort you devoted to reviewing our work.
> > >
> > > Thank you again!
> > >
> > > The authors

---

### Decision · Program_Chairs · 2026-04-30

**Decision:**

Accept (regular)

**Comment:**

This paper proposes a technique for improving the sampling quality of diffusion language models. The core idea is to rely on a recent buffer of historical predictions, instead of merely the current state, to better estimate the probability of the next step's tokens. Theoretical justifications is provided, and empirical evaluations show that the proposed method demonstrates improved performance, especially under highly parallel sampling settings. Strength wise, reviewers have found the method sound, with interesting theoretical analysis and reasonably good results. Initially, there are also several concerns over the clarify and several aspects of the evaluations. The authors had successfully resolved much of the concerns in the rebuttal, after which 3 reviewers are supportive, with 1 leaning towards reject (reviewer 8s5n). After the discussions, it seems like the main critique from reviewer 8s5n is the lack of accurate speed measure for comparisons. While this is a valid concern, the AC also believes it is on the minor side, and should not be a main obstacle from accepting this work. The AC suggests accepting this paper conditioned on the authors' promises of including all the changes mentioned in the rebuttal period.